# WAVY GROWTH Arabidopsis E3 ubiquitin ligases affect apical PIN sorting decisions

Nataliia Konstantinova[1,2,3], Lukas Hoermayer[4], Matouš Glanc[2,3,4], Rabab Keshkeih[1], Shutang Tan[4,6], Martin Di Donato[5], Katarzyna Retzer [1,7], Jeanette Moulinier-Anzola[1], Max Schwihla[1], Barbara Korbei [1], Markus Geisler [5], Jiří Friml [4] & Christian Luschnig [1] ✉

Directionality in the intercellular transport of the plant hormone auxin is determined by polar plasma membrane localization of PIN-FORMED (PIN) auxin transport proteins. However, apart from PIN phosphorylation at conserved motifs, no further determinants explicitly controlling polar PIN sorting decisions have been identified. Here we present Arabidopsis WAVY GROWTH 3 (WAV3) and closely related RING-finger E3 ubiquitin ligases, whose loss-of-function mutants show a striking apical-to-basal polarity switch in PIN2 localization in root meristem cells. WAV3 E3 ligases function as essential determinants for PIN polarity, acting independently from PINOID/WAG-dependent PIN phosphorylation. They antagonize ectopic deposition of de novo synthesized PIN proteins already immediately following completion of cell division, presumably via preventing PIN sorting into basal, ARF GEF-mediated trafficking. Our findings reveal an involvement of E3 ligases in the selective targeting of apically localized PINs in higher plants.

Morphogenesis and adaptive growth responses in plants are defined largely by directional, intercellular auxin transport and variations therein. Polar PIN protein localization at the plasma membrane (PM) establishes directionality of auxin flow to modulate developmental programs that rely on transiently formed auxin maxima and minima[1–4]. *Cis-* and *trans-*acting regulators of PIN sorting and polarity have been established in recent years, amongst which control by PINOID (PID)/WAG AGCVIII protein kinases appears unique, as they were found to control binary, apical-vs.-basal switches in PIN polarity[5]. Specifically, PID/WAGs-catalyzed PIN phosphorylation at conserved sites in the central cytoplasmic loop, favoring apical/shootward PIN localization via mechanisms that involve the control of lateral diffusion of PINs at the PM[5–7]. On the contrary, PIN dephosphorylation by antagonistically acting protein phosphatases coincides with PIN localization at the basal/rootward PM domain[7]. Phosphorylation-dependent PIN apical-vs.-basal polarity control, intimately interacts with ADP ribosylation factor guanine nucleotide exchange factor-(ARF GEF)-dependent cargo sorting, exemplified by Brefeldin A (BFA)-sensitive GNOM that is essential for PIN vesicular trafficking and intracellular recycling to the basal cellular PM domain[8,9]. Modes of action, by which PID/WAG and such ARF GEF pathways interact, and the nature of molecular switches involved in discriminating between apically and basally sorted PIN proteins, however, remain poorly characterized[10,11]. In addition, recent findings demonstrated that PIN phosphorylation at sites overlapping with those recognized by PID/WAGs, also functions in activating PIN-mediated cellular auxin efflux[12,13], without affecting protein polarity. This argues for a more complex

[1]Department of Applied Genetics and Cell Biology, Institute of Molecular Plant Biology, University of Natural Resources and Life Sciences, Vienna (BOKU), Muthgasse 18, 1190 Wien, Austria. [2]Department of Plant Biotechnology and Bioinformatics, Ghent University, 9052 Ghent, Belgium. [3]VIB Center for Plant Systems Biology, 9052 Ghent, Belgium. [4]Institute of Science and Technology Austria (IST Austria), 3400 Klosterneuburg, Austria. [5]Department of Biology, University of Fribourg, Chemin du Musée 10, 1700 Fribourg, Switzerland. [6]Present address: School of Life Sciences, Division of Life Sciences and Medicine, University of Science and Technology of China, Hefei 230026, China. [7]Present address: Institute of Experimental Botany of the Czech Academy of Sciences, Rozvojová 263, Praha 6, Czech Republic. ✉e-mail: christian.luschnig@boku.ac.at

regulation of PIN polarity acquisition and variations therein, and implies the involvement of additional mechanisms[6,14,15].

In this report, we describe *Arabidopsis WAVY GROWTH3* (*WAV3*) and *WAV3 HOMOLOG* (*WAVH*) RING-finger ubiquitin E3 ligases as mediators of PIN polarity. Together with their rice ortholog *SOIL SURFACE ROOTING1* (*SOR1*), these E3 ligases have originally been identified as regulators of root gravitropism and ethylene responses[16–18]. Here, we unravel a role for redundantly acting *WAV3/WAVH* genes in specifying apical-vs.-basal polar localization of PIN proteins in *Arabidopsis* root meristem cells. Genetic and cell biological analyses reveals a connection between E3 ubiquitin ligase function and the control of apical PIN cargo sorting, influencing directional auxin flow and root growth.

## Results

### WAV3/WAVH in conjunction with PIN2 regulate shootward auxin transport in roots

*WAV3*, and closely related *WAVH1* and *WAVH2* have originally been identified as redundantly acting regulators of root gravitropism, reflected in additive directional root growth defects in *wav3 wavh1 wavh2* (here, *wav triple*) loss-of-function mutants[16]. Furthermore, altered auxin signaling of *wav triple* suggested roles for these E3 ligases in modulating auxin-controlled gravitropic root bending[16]. *pin2* alleles, deficient in shootward auxin transport from the root tip into the root elongation zone, resemble *wav triple* as they also exhibit root gravitropism defects, altered ethylene responsiveness, and ectopic auxin distribution in root meristems[19,20]. This prompted us to test genetic interaction by generating a *wav3-1 wavh1-1 wavh2-1 eir1-4* (a *pin2* null allele[21]) quadruple mutant. Roots of this line exhibited root gravitropism defects and increased resistance to the ethylene precursor ACC (1-aminocyclopropane-1-carboxylic acid), highly similar to *eir1-4* and *wav triple* parental lines (Fig. 1a; Supplementary Fig. 1a). These apparently non-additive effects on root growth hint at epistatic interactions, in which activities of *PIN2* and *WAV3/WAVH* are jointly required for coordinating auxin effects on root growth.

Expression of *WAV3/WAVH* translational reporter genes fused to the Venus tag overlaps with *PIN2* expression in the root meristem. *WAVH1::WAVH1:Ven* reporter signals were observed in stele and root cap cells, *WAV3::Ven:WAV3* signals predominantly in the epidermis of the cell division zone, whereas *WAVH2::WAVH2:Ven* expression peaked in the epidermis of the root elongation zone (Fig. 1b–f). This pattern is in agreement with crosstalk between these E3s and PIN2, but also suggests spatially discernible functions of *WAV3/WAVH* genes. Support for this comes from observations demonstrating that *WAV3::Ven:WAV3* expression causes only limited rescue of *wav triple* root phenotypes (Fig. 1g). In contrast, ectopic expression of *WAV3* in the entire root meristem mediated by estradiol-inducible *XVE»Venus:WAV3*, resulted in efficient rescue of *wav triple* root elongation and gravitropism defects (Fig. 1a, f, g; Supplementary Fig. 1b–e), indicating functionally equivalent roles for *WAV3* and *WAVH* genes in overlapping root meristem cell files.

To assess auxin distribution in *wav triple* we introduced and analyzed auxin-responsive *DR5rev:3XVENUS-N7*[22]. These experiments revealed ectopic reporter expression in distal portions of *wav triple* root meristems in proximity of the stem cell niche, an expression pattern consistent with increased auxin accumulation due to defects in shootward auxin transport (Fig. 1h, i). Similar observations were made with *wav triple XVE»Ven:WAV3 DR5rev:3XVENUS-N7* seedlings, with induction of Venus-WAV3 expression causing a disappearance of DR5-Venus signals in lateral portions of primary root meristems (Supplementary Fig. 2a–d), and indicating that loss of *WAV3/WAVH* genes interferes with auxin distribution or signaling in root meristems. We therefore made use of the conditionally complementing *wav triple XVE»Ven:WAV3* line for determining the shootward relocation of $^3$H-labeled IAA from the root tip into the root elongation zone. No

significant differences in shootward auxin tracer relocation were observed, when comparing wild type and *wav triple XVE»Ven:WAV3* grown in presence of estradiol (Fig. 1j). However, washout and depletion of estradiol for 6 h, which resulted in diminished reporter signals and root gravitropism defects without affecting the overall root morphology (Supplementary Fig. 2e–g), caused a significant reduction in $^3$H-IAA relocation in *wav triple XVE»Ven:WAV3* root meristems in comparison to wild type (Fig. 1j). As internal control, we simultaneously determined shootward redistribution of $^{14}$C-benzoic acid (BA) as a diffusion control over time, which revealed no differences between wild type and *wav triple XVE»Venus:WAV3* root meristems (Fig. 1j). Furthermore, control assays, employing the very same experimental set-up with the shootward auxin transport-deficient *eir1-4* allele, revealed no effects of the conditions used on radioactive tracer distribution (Supplementary Fig. 2h). The observed defects in $^3$H-IAA relocation in *wav triple* thus link the function of WAV3 to shootward auxin transport from the root tip into the root elongation zone, similar as known for PIN2[20].

In summary, the genetic interaction, overlapping expression, and related functions in mediating shootward auxin transport strongly suggest that PIN2 and WAV3 act in conjunction.

### WAV3/WAVH genes are required for apical vs. basal PIN targeting

*Wav triple* deficiencies in shootward auxin transport are suggestive of WAV3/WAVH activities in controlling some aspect of PIN2 activity; e.g., abundance or localization. Nonetheless, quantification of PIN2 protein levels in *wav triple* revealed no differences, indicating that the E3s are dispensable for PIN2 turnover (Supplementary Fig. 3a). We then determined PIN2 localization by immuno-staining and detected a binary polarity shift in protein distribution in *wav triple*. Unlike wild type, in which PIN2 localizes to the apical PM domain of lateral root cap (LRC), epidermis and elongating cortex cells, but basally in the lower part of the cortex cell files[23], *wav triple* root meristems exhibited a strictly basal PM localization of PIN2 in all cell files (Fig. 2a–d). This ectopic PIN2 localization is completely reverted by *XVE»Ven:WAV3* expression in *wav triple*, confirming that loss of *WAV3/WAVH* is causing PIN polarity defects (Supplementary Fig. 3b). Apical PIN targeting different from PIN2 in *wav triple* was assessed further by employing two different PIN1-GFP translational fusions, expressed under control of the *PIN2* promoter, which show opposite cellular polarity in root epidermal cells, presumably as a result of different protein conformation due to a differing positioning of the GFP tag within the central hydrophilic loop of PIN1[23]. Signals of the one PIN1-GFP reporter (*PIN2::PIN1:GFP-2*) in root meristem cells were exclusively found at the basal PM domain, both in *eir1-1* and in *wav triple*, indicating no interference with basal sorting of ectopically expressed PIN1 (Fig. 2e, f; Supplementary Fig. 3c). The other reporter, *PIN2::PIN1:GFP-3*, exhibited a predominantly apical reporter protein distribution in *eir1-1*, reminiscent of endogenous PIN2 localization and sufficient to rescue *eir1-1* root gravitropism defects (Fig. 2g)[23]. In contrast, in *wav triple*, PIN1:GFP-3 located exclusively to the basal PM domain, coinciding with strong defects in gravitropic root growth (Fig. 2h; Supplementary Fig. 3c). Finally, when determining the localization of basally localized PIN1 in root stele cells, no aberrations in basal PM localization were observed in *wav triple*, substantiating the notion that WAV3/WAVH are essential for the apical localization of PINs, whilst their basal sorting appears unaffected (Fig. 2i, j).

To characterize further the specificity of WAV3/WAVH in PM protein sorting, we analyzed localization of additional polarly targeted PM reporter proteins in *wav triple* root meristem cells. These involved proteins, demonstrated to be enriched at apical or apolar (*OCTOPUS, OPS::OPS:mCit*[24]; *AUXIN RESISTANT 1; AUX1::YFP:AUX1*[25]), basal (*D6PK::YFP:D6PK*[26]) and lateral PM

 

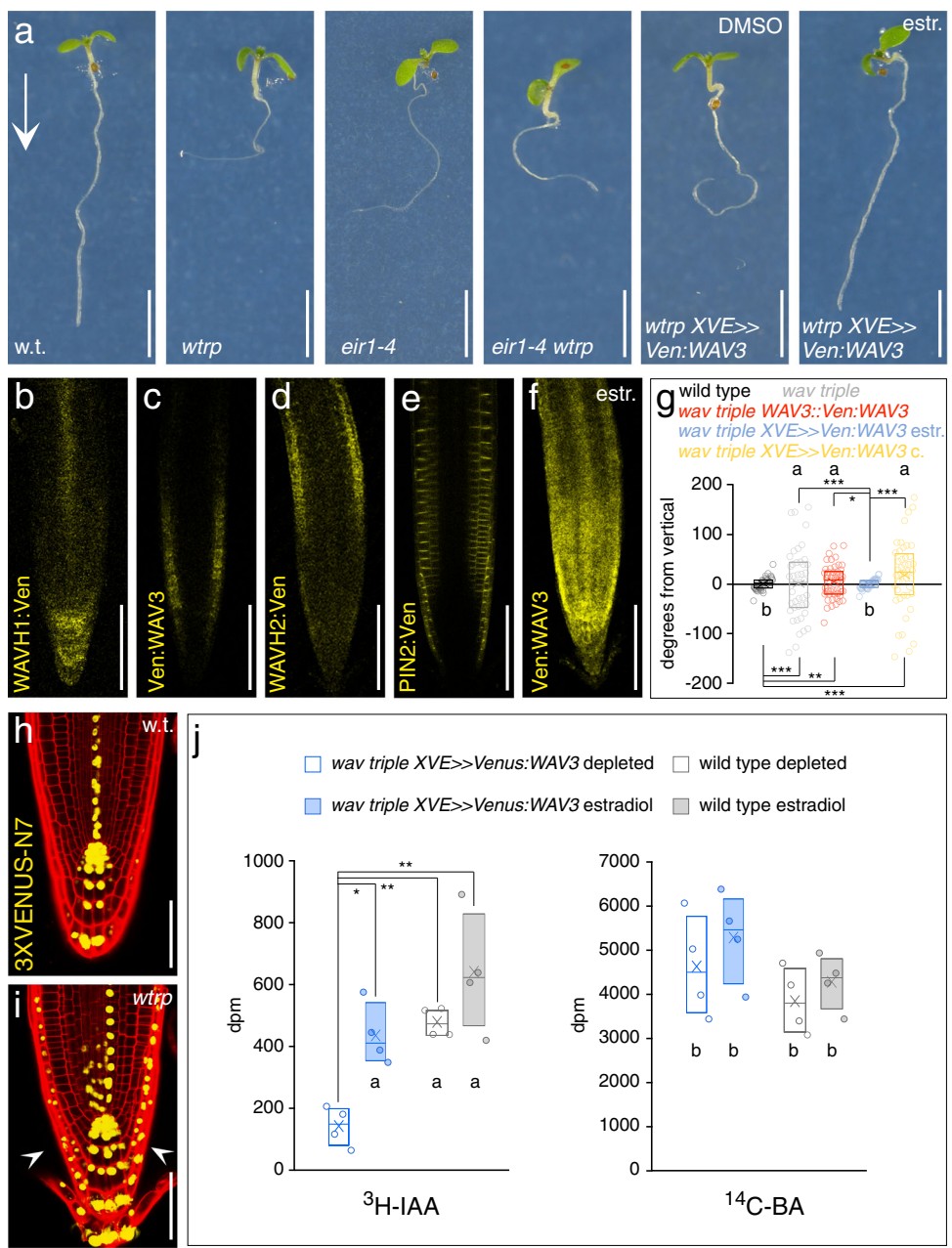

**Fig. 1 | Arabidopsis WAV3/WAHV genes act in root gravitropism and polar auxin transport in root meristems. a** Comparison of wild type (w.t.), *wav triple* (*wtrp*), *eir1-4*, *eir1-4 wav triple*, *wav triple XVE»Ven:WAV3* in presence of solvent (DMSO) or 2 µM estradiol. 5 DAG seedlings grown on vertically oriented plates are on display. Arrow indicates gravity vector. Reporter signals in *WAHV1::WAVH1:Ven* (**b**), *wav triple WAV3::Ven:WAV3* (**c**), *WAHV2::WAVH2:Ven* (**d**), *eir1-4 PIN2::PIN2:VEN* (**e**) and *wav triple XVE»Ven:WAV3* in presence of 2 µM estradiol (**f**). Roots of 5 DAG seedlings are on display. **g** Orientation of root tips, expressed as degrees deviation from vertical of 5 DAG wild type (*n* = 46 roots/3 experiments), *wav triple* (*n* = 44 roots/3 experiments), *wav triple WAV3::Ven:WAV3* (*n* = 52 roots/3 experiments), *wav triple XVE»Ven:WAV3* seedlings grown on solvent (DMSO, 'c.'; *n* = 43 roots/3 experiments) or in presence of 2 µM estradiol ('estr.'; *n* = 54 roots/3 experiments). Expression of *DRSrev:3XVENUS-N7* in wild type (w.t.; **h**) and *wav triple* (*wtrp*; **i**) root meristems at 5

DAG. Arrowheads indicate ectopic reporter signals. **j** Root shootward (basipetal) PAT measurement using ³H-IAA and as a control, ¹⁴C-benzoic acid (BA, diffusion control) performed with 5 DAG wild type and *wav triple XVE»Ven:WAV3* seedlings. Radiotracers were applied to the root tip followed by their quantification 5 mm above the very root tip. Assays were conducted in presence of 2 µM 17β-estradiol ('estradiol') or in presence of DMSO solvent only ('depleted'); for further details see Methods (*n* = 4, with 20 seedlings each). Three independent experiments were performed for (**a**), (**b**–**f**), and (**h**, **i**) with similar results. Circles represent data points; boxes: first and third quartiles; center line: median; 'x': mean value. Levene's tests were employed to determine the equality of variances (**g**); One-way ANOVA with post hoc Tukey HSD was performed (**j**); *: $p < 0.05$; **: $p < 0.01$; ***: $p < 0.001$; **a**, **b**: $p > 0.05$. Scale bars: *a* = 10 mm; **b**–**f** = 50 µm; **h**, **i** = 25 µm. Source data are provided as Source Data file.

domains (BORON TRANSPORTER 1, *BOR1::BOR1:GFP*[27]; PENETRATION 3, *PEN3::PEN3:GFP*[28]) but showed no altered localization in *wav triple* (Fig. 3a–e). Furthermore, to determine the consequences of *wav triple* on apical plasma membrane distribution specifically in the *PIN2* expression domain in root meristems, we conditionally expressed the apical marker OPL2:sYFP (OCTOPUS-LIKE 2[29]) via

*PIN2*-promoter controlled expression of the *XVE* transcriptional regulator. Analysis of wild type and *wav triple* seedlings expressing *PIN2::XVE»OPL2:sYFP* exhibited reporter signals predominantly at the apical domain of root meristem cells (Fig. 3f), demonstrating that WAV3/WAVH modulate targeting of only a subset of apically localized PM proteins, exemplified by PIN proteins.

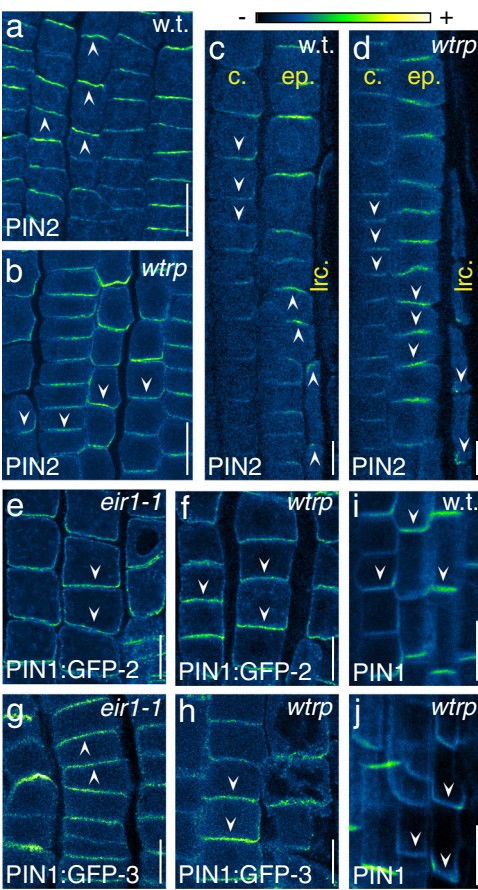

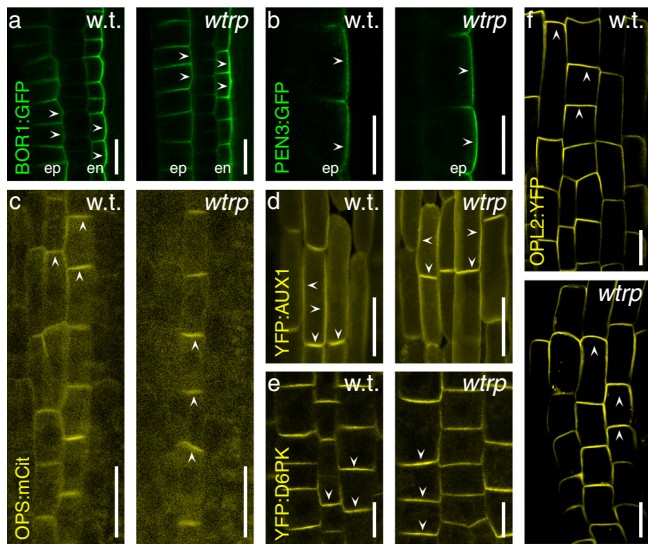

**Fig. 3 | Subcellular localization of polarly localized PM-associated reporter proteins in 5 DAG wild type (w.t.) and *wav triple* (*wtrp*). a** *BOR1::BOR1:GFP* in root meristem epidermis ('ep') and endodermis ('en') cells. **b** *PEN3::PEN3:GFP* in root meristem epidermis cells. **c** *OPS::OPS:mCitrine* in root meristem stele cells. **d** *AUX1::YFP:AUX1* in root meristem LRC cells. **e** *D6PK::YFP:D6PK* in root meristem epidermis cells. **f** *PIN2::XVE»OPL2:YFP* in root meristem epidermis cells. Three independent experiments were performed for (**a**–**f**) with similar results. Arrowheads indicate membrane localization of reporter protein signals. Size bars: **a**–**f** = 10 μm.

**Fig. 2 | PIN polarity switches in *wav triple* root meristems.** PIN2 immunostaining in wild type (w.t.; **a**, **c**) and *wav triple* (*wtrp*; **b**, **d**) root meristems at 5 DAG. **a**, **b** Epidermis cells; **c**, **d** cortex ('c.'), epidermis ('ep.'), and LRC ('lrc.') cells. Immunostaining of *PIN2::PIN1:GFP-2* (**e**, **f**) and *PIN2::PIN1:GFP-3* (**g**, **h**) reporter proteins in 5 DAG *eir1-1* (**e**, **g**) and *wav triple* (**f**, **h**) root meristem epidermis cells. PIN1 immunostaining in 5 DAG wild type (w.t.; **i**) and *wav triple* (*wtrp*; **j**) root stele cells. Three independent experiments were performed for (**a**–**j**) with similar results. Arrowheads indicate the polarity of PIN signals. Scale bars: **a**, **b** = 20 μm; **c**–**j** = 10 μm.

## WAV3/WAVH act interdependently with *PID/WAG* and antagonize ARF GEF-dependent PIN sorting

Our experiments identified WAV3/WAVH E3 ubiquitin ligases as unique mediators of PIN polarity. Therefore, and since PIN2 has been found to undergo covalent modification by ubiquitin to modulate its sorting and vacuolar degradation[30,31], we tested for effects of PIN2 ubiquitylation on its polar distribution. However, neither a PIN2-ubiquitin fusion protein mimicking constitutive PIN2 ubiquitylation, nor pin2[K12R] deficient in ubiquitylation[30], exhibited altered polarity when compared to wild type PIN2 localization (Supplementary Fig. 4a–e). Moreover, both *pin2* ubiquitylation alleles did not revert PIN2 polarity defects, when expressed in *wav triple* (Supplementary Fig. 4a–e), indicating that manipulation of the PIN2 ubiquitylation status does not affect its subcellular polarity or WAV3/WAVH action on it.

Phosphorylation of conserved motifs within the hydrophilic PIN loop by the PID/WAG AGCVIII Ser/Thr protein kinases, on the other hand, promotes apical targeting of PINs[7,32,33], similar to what we attribute to WAV3/WAVH. Consistent with such activities, a loss of *PID/WAG* genes causes basal distribution of otherwise apically localized PIN proteins, not as pronounced but highly similar to *wav triple* phenotypes, whereas *PID/WAG* overexpression results in apical PIN localization[5,33]. We tested for interaction between PID/WAGs and WAV3/WAVHs by analysis of loss-of-function mutant combinations.

*pid-14* loss-of-function phenotypes are characterized by deviations in cotyledon numbers and naked pin-like inflorescences, enhanced further in *pid-14 wag1-1 wag2-1*, which exhibits additional root gravitropism defects and a complete deficiency in cotyledon formation[33]. Remarkably, we observed similar phenotypes, when introducing only *pid-14* into *wav triple*. Apart from the characteristic *pid* inflorescence phenotypes and *wav triple* defects in root growth, *wav triple pid-14* also developed cotyledon-less seedlings, which we never observed in the parental lines (Fig. 4a–g; Supplementary Table 1). *Wav triple pid-14 wag2-1* lines exhibited even stronger phenotypes. This pentuple mutant formed cotyledon-less seedlings and frequently exhibited deficiencies in shoot differentiation, preventing mutant individuals to develop beyond early developmental stages and causing arrested growth as irregularly shaped seedlings (Fig. 4h–m; Supplementary Table 1). Those plantlets that escaped such growth arrest exhibited strong limitations in shoot development, substantially more pronounced than in *pid-14* and *wag1-1 wag2-1 pid-14* (Supplementary Fig. 5a). Overall, our observations suggest additive effects of *wav3/wavh* and *pid/wag* mutants already during embryogenesis. Consistently, attempts to obtain *wav triple pid-14 wag1-1* and *wav triple pid-14 wag1-1 wag2-1* combinations failed, implying that these mutant combinations are no longer viable.

Evidence for interdependent, synergistic roles of WAV3/WAVHs and PID/WAGs, was obtained by introducing *35S::PID* into *wav triple*. Increased PID levels in *35S::PID* causes strict apicalization of all PINs in root meristem cells, severely interfering with directional auxin flow to the root tip, causing root agravitropism and root meristem consumption[5]. *wav triple 35S::PID* roots still exhibited agravitropic growth, but root meristem consumption was no longer observed, suggesting that *PID* overexpression-induced defects in polar auxin transport are alleviated by *wav triple* (Supplementary Fig. 5b–f). In line with this assumption, PIN2 localization turned out to be intermediate in *wav triple 35S::PID*, illustrated by cells and cell files exhibiting mixed apical, basal, or bipolar apical/basal protein localization at the PM (Fig. 5a–f). Similar observations were made for PIN1, with *wav triple*

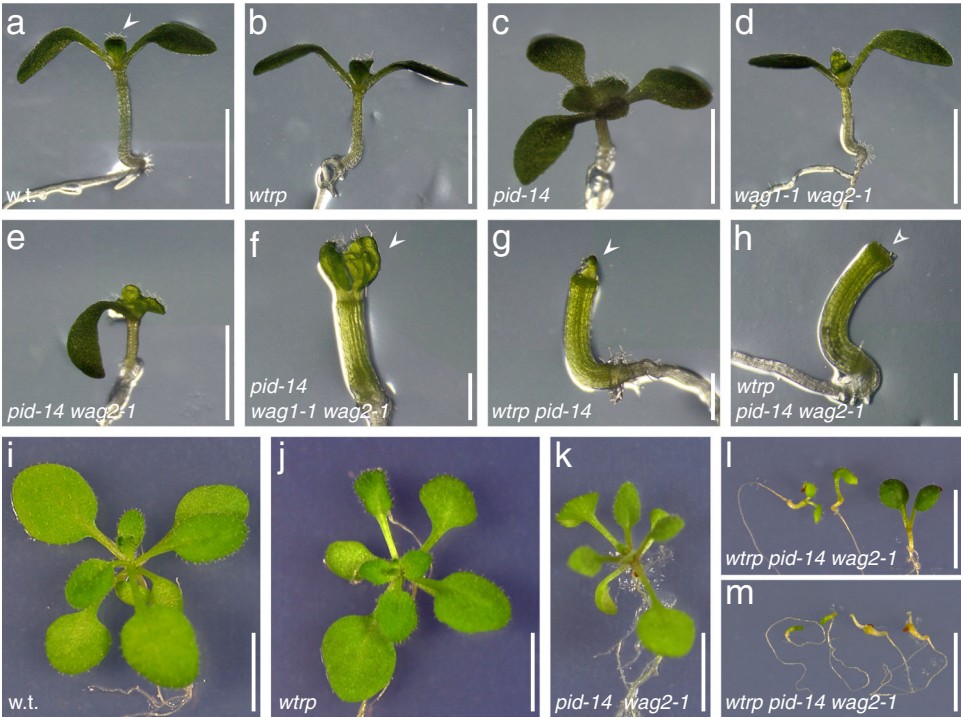

**Fig. 4 | Genetic interaction between *PID/WAG* and *WAV3/WAVH* genes in the regulation of PIN2 polarity.** Wild type (w.t.; **a**), *wav triple* (*wtrp*; **b**), *pid-14* (**c**), *wag1-1 wag2-1* (**d**), *pid-14 wag2-1* (**e**), *pid-14 wag1-1 wag2-1* (**f**), *wav triple pid-14* (**g**), *wav triple pid-14 wag2-1* (**h**) hypocotyl and cotyledon phenotypes of seedlings, 5 DAG. Arrowheads indicate emerging true leaves (**a**, **f**, **g**), which cannot be observed in *wav triple pid-14 wag2-1* at this stage (h, open arrowhead). Wild type (w.t.; **i**), *wav triple* (*wtrp*, **j**), *pid-14 wag2-1* (**k**), *wav triple pid-14 wag2-1* (**l**, moderate phenotypes; **m**, severe phenotypes) at 16 DAG. Scale bars: **a**–**e** = 3 mm; **f**–**h** = 1 mm; **i**–**m** = 10 mm.

partially reverting apical PIN1 localization in *35S::PID* root stele cells, presumably participating in re-establishment of rootward auxin flow thereby antagonizing root growth arrest (Supplementary Fig. 5g–j).

To further elaborate, whether PIN2 phosphorylation could overrule PIN2 mislocalization in *wav triple*, we employed *PIN2::pin2^{S1,2,3D}:Dendra*, in which serine at positions 237, 258 and 310 is replaced by aspartic acid, to produce a *pin2* allele mimicking phosphorylation at sites recognized by PID/WAGs[34]. Comparison of protein localization in *eir1-4 PIN2::PIN2:Dendra* and *eir1-4 PIN2::pin2^{S1,2,3D}:Dendra* root meristem cells, revealed apical localization in epidermis cells of both reporter lines. In cortex cells of the cell division zone, the basal PIN2:Dendra localization resembled wild type PIN2, however, pin2^{S1,2,3D}:Dendra signals showed pronounced apical localization (Fig. 5g–k). This mislocalization is reminiscent of ectopic distribution of phosphomimic pin1^{S1,2,3E} in root stele cells[32], implying that mimicking PID/WAG-mediated PIN2 phosphorylation promotes apicalization of PIN2. In *wav triple eir1-4 PIN2::PIN2:Dendra* we observed a uniform basal localization of the reporter protein, indistinguishable from the mislocalization of endogenous PIN2 in *wav triple*. In contrast, *wav triple eir1-4 PIN2::pin2^{S1,2,3D}:Dendra* exhibited a mixed distribution of reporter signals with basal, apical, and bipolar distribution in root meristem cells (Fig. 5i–k). This intermediate distribution argues for antagonistic effects of mimicking PID/WAG-mediated PIN2 phosphorylation, which promotes apical protein distribution, and loss of *WAV3/WAVH*, which causes basal localization. In further experiments, we asked if a loss of *WAV3/WAVH* genes might impact on steady-state protein levels or subcellular distribution of PID and expressed the *PID::PID:YFP* reporter gene in *wav triple*. No such differences were observed, when comparing *wav triple* and wild type controls, suggesting that WAV3/WAVH E3 ubiquitin ligases activity do not impact on PID abundance and/or distribution (Supplementary Fig. 6a–c).

Overall, our analysis of WAV3/WAVH E3s and PID/WAG protein kinases revealed separable roles. Evidently, both function in the control of apical-vs.-basal targeting of PIN proteins, and crosstalk between these classes of proteins appear to occur in all likelihood. The lack of evidence for clear-cut epistatic interactions, however, implies distinct modes of action, by which these proteins control polar PIN distribution.

Accumulation of PINs particularly at the basal PM domain requires activity of BFA-sensitive ARF GEFs[8,10]. This is underlined by observations demonstrating that PIN sorting and recycling to the basal PM domain is substantially more sensitive to BFA treatment than sorting to apical PM domains. As a result, such inhibition causes a preferential retention of cargo destined for the basal PM domain in BFA-induced aggregated endosomal compartments[35]. Quantification of reporter signals in BFA-treated seedlings demonstrated a significant signal increase in BFA compartments in *wav triple PIN2::PIN2:VEN* root epidermis cells, when compared to controls, indicating that basal PIN2 accumulation in *wav triple* occurs via BFA-sensitive pathways (Fig. 6a). To dissect which specific trafficking pathway(s) are most affected in *wav triple*, we then determined relative PIN2-Venus abundance in BFA compartments upon inhibiting translation, and thus preventing the secretion of newly synthesized protein from contributing to BFA body formation. In *eir1-4 PIN2::PIN2:VEN* seedlings, with apically localized PIN2, cycloheximide (CHX) pretreatment followed by CHX/BFA co-incubation did not result in notable effects on signal ratios. This suggests that BFA body-retained PIN2-Venus comes predominantly from the recycling pathway, in line with a limited BFA responsiveness of de novo secretion of apically targeted proteins such as PIN2[35]. In contrast, *wav triple PIN2::PIN2:VEN* seedlings responded to CHX pre-treatment followed by CHX/BFA co-incubation with a markedly decreased plasma membrane-to-BFA compartment ratio. This indicates that in the *wav triple* mutant, much of the basal PM-localized PIN2-Venus is of de novo secretory origin, while endocytic PIN2-Venus sorting from the basal plasma membrane domain is still operational in the mutant, (Supplementary Fig. S7a–e). Collectively, these results indicate that WAV3/WAVH function predominantly in the sorting of newly synthesized PIN2 into the apical secretory pathway. Next, we determined

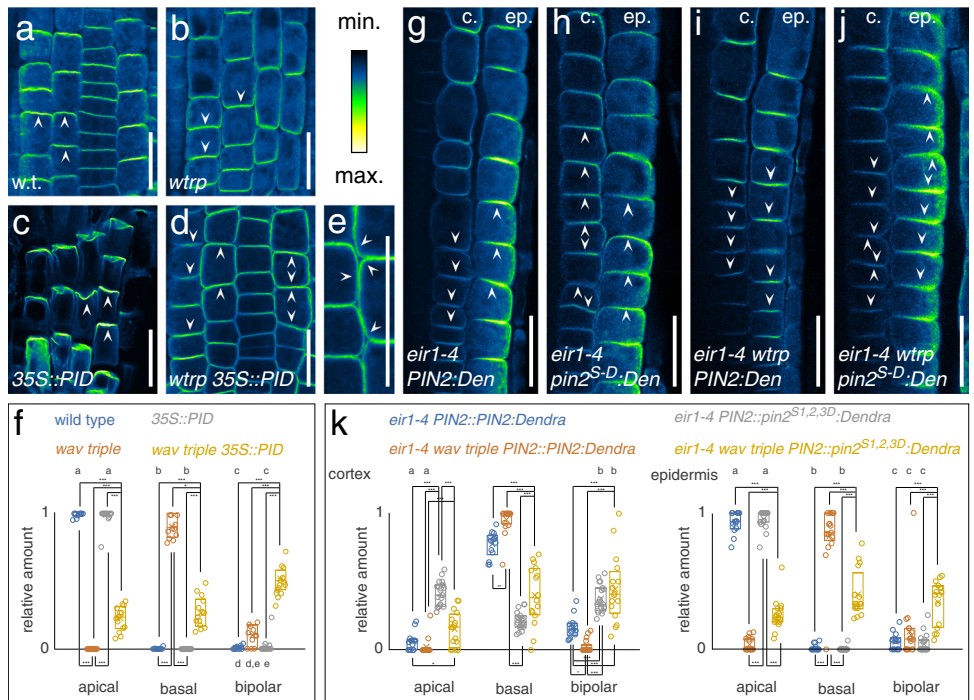

**Fig. 5 | Interplay of PINOID/WAG activity and *WAV3/WAVH* genes in PIN2 polarity control.** PIN2 localization in 4 DAG wild type (w.t.; **a**, **f**; 359 cells/9 roots), *wav triple* (*wtrp*; **b**, **f**; 265 cells/11 roots), *35S::PID* (**c**, **f**; 317 cells/15 roots) and *wav triple 35S::PID* (**d**–**f**; 429 cells/14 roots) root meristem epidermis cells. Arrowheads indicated polarity of PIN2 localization. **g**–**k** PIN2 localization in cortex ('c.') and epidermis ('ep.') root meristem cells of *eir1-4 PIN2::PIN2:Dendra* (*eir1-4 PIN2:Den*; **g**, **k**; 243 cortex cells/212 epidermis cells/14 roots), *eir1-4 PIN2::pin2^{S1,2,3D}:Dendra* (*eir1-4 pin2^{S-D}:Den*; **h**, **k**; 326 cortex cells/372 epidermis

cells/19 roots), *eir1-4 wav triple PIN2::PIN2:Dendra* (*eir1-4 wtrp PIN2:Den*; **i**, **k**; 193 cortex cells/210 epidermis cells/17 roots) and *wav triple eir1-4 PIN2::pin2^{S1,2,3D}: Dendra* (*eir1-4 wtrp pin2^{S-D}:Den*; **j**,**k**; 205 cortex cells/214 epidermis cells/16 roots) at 5 DAG. Arrowheads indicated polarity of PIN2 localization. Scale bars: **a**–**e**, **g**–**j** = 20 μm. Circles represent data sets from individual root meristems; boxes: first and third quartiles; center line: median; '*x*': mean value. One-way ANOVA with post hoc Tukey HSD was performed; *: $p < 0.05$; ***: $p < 0.001$; **a**–**e**: $p > 0.05$. Source data are provided as Source Data file.

---

consequences of prolonged BFA treatment, as such selective inhibition of protein sorting was found to cause apical localization of otherwise basally localized PINs[10,12], and we observed a comparable apical relocation of PIN2 in *wav triple* (Fig. 6b, c). Likewise, long-term treatment of *wav triple* seedlings with low BFA concentrations resulted in apical PIN2 localization, which coincided with a restoration of gravitropic root growth (Fig. 6d; Supplementary Fig. 8a–d). Strikingly, in *eir1-4 wav triple* we failed to observe BFA-induced restoration of directional root growth, indicating that apical relocation of PIN2, specifically mediates this response to BFA (Fig. 6d). Amongst ARF GEFs, GNOM function is essential for PIN localization at basal PM domains, and we therefore determined effects of the partial loss-of-function *gnom^{R5}* allele in *wav triple*[9]. Analogous to extended BFA treatment, gravitropic root growth is partially restored in *wav triple gnom^{R5}* seedlings and we observed reestablishment of apical PIN2 localization in root meristem epidermis cells (Fig. 6e–i). This demonstrates that loss of *WAV3/WAVH* causes PIN2 rerouting into GNOM-dependent sorting pathways that act in maintaining basal distribution of PINs. Nevertheless, such erroneous PIN rerouting does not coincide with altered abundance or subcellular distribution of GNOM, indicated by unaffected expression of a *GNOM:GFP* reporter gene in *wav triple* (Supplementary Fig. 8e–g).

Taken together, the restoration of PIN2 apical localization upon inhibition of ARF GEF function in *wav triple*, implies that loss of WAV3/WAVH E3s does not interfere with the overall functionality of the apical PIN sorting machinery, but causes erroneous entry into basal cargo sorting pathways. Furthermore, the coinciding restoration of apical PIN2 localization and gravitropic root growth upon preventing basal PIN sorting in *wav triple*, strongly supports a scenario, in which basal mislocalization of PIN2 represents a primary cause for the mutant's root gravitropism defects.

## WAV3/WAVH influence PIN2 polarity establishment following completion of cytokinesis

We analyzed subcellular localization of Venus-tagged *WAV3/WAVH* translational *Arabidopsis* reporter lines to gain more insights into the role of these E3 ligases. *wav triple WAV3::Ven:WAV3, wav triple XVE»Ven:WAV3, WAVH1::WAVH1:Ven*, and *WAVH2::WAVH2:Ven* lines, all exhibited polar PM-associated signals. The intensity of reporter protein signals peaked at the basal PM domain, with weaker signals found at lateral domains of root meristem cells. Additional, but less pronounced signals were also observed in cytoplasm and nucleus, indicative of distinct functions for the E3 ligases in different compartments (Fig. 7a–d; Supplementary Fig. 9a). In agreement, Chen and colleagues described the putative rice *WAV3/WAVH* ortholog *SOR1*, as a regulator of Aux/IAA protein stability, likely acting in the cell's interior[17]. We asked if *SOR1* represents a functional orthologue of Arabidopsis *WAV3/WAVH* genes in the control of PIN polarity and expressed a *GFP:SOR1* fusion gene under control of the CaMV 35S promoter in *wav triple*. Resulting lines exhibited rescue of *wav triple* root growth defects and a subcellular GFP:SOR1 reporter protein localization highly reminiscent of WAV3/WAVH reporter proteins (Fig. 7e; Supplementary Fig. 9b, c). Moreover, apical localization of PIN2 was restored in *wav triple 35S::GFP:SOR1* root meristem cells, demonstrating that the rice gene functions in apical PIN polarity control (Fig. 7f). In additional experiments we asked if ectopic *WAV3* expression in wild type might impact on PIN2 polar localization. Expression of either *WAV3::Ven:WAV3* in wild type, produced a reporter signal distribution similar to the one observed in *wav triple* (Supplementary Fig. 10a, b), and such basal WAV3 reporter protein localization was also observed in wild type expressing *XVE»Ven:WAV3* (Supplementary Fig. 10c). Assessment of subcellular PIN2

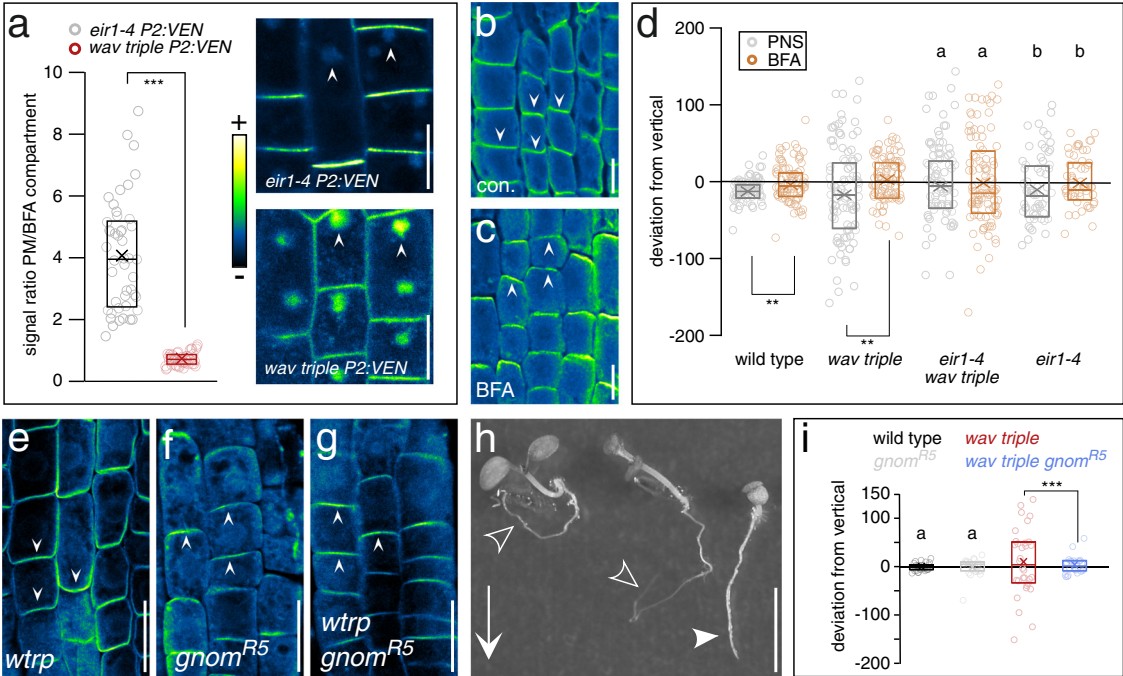

**Fig. 6 | Wav triple-induced PIN2 polarity switches are mediated by BFA-sensitive cargo sorting. a** *PIN2::PIN2:VEN* reporter signal distribution in root meristem epidermis cells of 6 DAG *eir1-4* (*n* = 47 cells/7 roots) and *wav triple* (*n* = 46 cells/8 roots) incubated in presence of 50 µM BFA for 90 min. Arrowheads indicate accumulation of reporter signals in BFA-induced intracellular compartments. **b, c** Subcellular PIN2 localization in *wav triple* root meristem epidermis cells in 6 DAG seedlings incubated on control plates ('con.'; **b**) or in presence of 50 µM BFA ('BFA'; **c**) for 16 h. Arrowheads indicate PIN2 polarity. **d** Orientation of root tips, expressed as degrees deviation from vertical of 6 DAG seedlings germinated on nutrient plates and then transferred on control nutrient plates (*n* = 81, wild type; *n* = 99, *wav triple*; *n* = 77, *eir1-4 wav triple*; *n* = 60, *eir1-4*) or on plates supplemented with 10 µM BFA (*n* = 85, wild type; *n* = 96, *wav triple*; *n* = 101, *eir1-4 wav triple*; *n* = 54, *eir1-4*) for 36 h. Subcellular localization of PIN2 in root meristem epidermis cells of 6 DAG *wav triple* (**e**; *wtrp*), *gnom^{R5}* (**f**) and *wav triple gnom^{R5}* (**g**). Arrowheads indicate PIN2 polarity. **h** Segregating *wav triple* and *wav triple gnom^{R5}* seedlings cultivated on vertically oriented nutrient plates. Open arrowheads indicate agravitropic *wav triple* primary roots, closed arrowhead indicates root of *wav triple gnom^{R5}*. Arrow indicates the gravity vector. **i** Orientation of root tips, expressed as degrees deviation from vertical of 8 DAG wild type (*n* = 30), *wav triple* (*n* = 32), *gnom^{R5}* (*n* = 24) and *wav triple gnom^{R5}* (*n* = 26) seedlings cultivated on vertically oriented nutrient plates. Three independent experiments were performed for (**b, c**) and (**e–g**) with similar results. Circles represent single data points; boxes: first and third quartiles; center line: median; 'x': mean value. Two-tailed t-test (**a, d**) and Levene's tests for the equality of variances (**i**) were employed to determine statistical significance; **: $p < 0.01$; ***: $p < 0.001$; **a, b**: $p > 0.05$. Scale bars: **a–c** = 10 µm; **e–g** = 20 µm; **h** = 10 mm. Source data are provided as Source Data file.

localization in wild type root meristem cells, conditionally expressing *XVE»Ven:WAV3* revealed no striking differences to controls, implying that conditional expression of an extra copy of *WAV3* in wild type does not result in dosage-dependent effects on the localization of PIN2 (Supplementary Fig. 10d, e).

In interphase cells we failed to observe well-defined overlaps in the localization of PM-associated WAV3/SOR1 reporter proteins, which are enriched at the basal PM domain, and PIN2, which for the most part locates to the apical PM domain (Fig. 7h). When viewing cytokinetic root meristem cells we detected Venus:WAV3 in proximity of the cell plate (Fig. 7g, i), which might point towards WAV3 functions in cargo distribution control already upon completion of cell division. Therefore, and to obtain insights into WAV3 activities in cytokinetic cells, we analyzed cell plate-localized PIN2 sorting kinetics in more detail (Fig. 7i). For this purpose, we made use of the *KN::PIN2:Dendra* reporter gene, expressing *PIN2* under control of the cytokinesis-specific *KNOLLE* promoter[36]. In wild type, *KN::PIN2:Dendra* signals were demonstrated to build up at the cell plate, which in the apical cell is followed by PIN2 accumulation at the apical PM domain, restoring polar protein localization upon completion of cell division. In the basal cell, PIN2 at the cell plate defines the newly established apical domain of this daughter cell (Fig. 8a, b, d)[37]. In *wav triple KN::PIN2:Dendra* we also detected reporter protein accumulation at the cell plate of dividing cells. However, PIN2:Dendra also accumulated at the basal pole of the basally localized daughter cell, whereas in the apically localized cell, the reporter protein for the most part locates to the cell plate and failed to accumulate at the apical PM domain (Fig. 8a, b, d).

*wav triple* therefore exhibits PIN2 polarity establishment defects already in cytokinetic cells.

Since WAV3/WAVH/SOR1 encode for E3s, we asked if the correct PIN2 polarity may require proteasome activity. Treatment with proteasome inhibitor MG132 caused increased intracellular signal accumulation in *KN::PIN2-Dendra* lines, but we still observed apical PIN2 localization in wild type and basal accumulation in *wav triple*, and similar findings were made for endogenous PIN2 in MG132-treated wild type and *wav triple XVE»Ven:WAV3* (Fig. 8c and Supplementary Fig. 11a–d). This implies involvement of proteasome activity in PIN2-sorting, but no specific effects on its polar distribution that might account for PIN2 polarity defects in *wav triple*.

To follow the kinetics of PIN2 sorting in cytokinetic root meristem cells, we photo-converted the pre-existing PIN2:Dendra protein pool which allowed us to determine the fate of PIN2 synthesized de novo. In wild type, such PIN2 is sorted to the apical PM domain in the apical cell, whereas in the basal cell, most of de novo synthesized protein accumulates at the cell plate, which subsequently defines the apical domain of the basal cell (Fig. 8f; Supplementary Fig. 11e). In *wav triple*, de novo synthesized PIN2 signals are pronounced at the basal pole of the basally localized daughter cell, demonstrating active missorting of PIN2. In the apical cell, de novo synthesized PIN2:Dendra failed to get efficiently sorted to the apical domain, likewise reflecting erroneous PIN2 targeting (Fig. 8f). Finally, we determined if PIN2 misrouting in cytokinetic cells depends on BFA-sensitive ARF GEFs, and treated *KN::PIN2:Dendra* lines with low concentrations of BFA. In wild type, we observed PIN2 localization primarily at the apical domain as well as cell

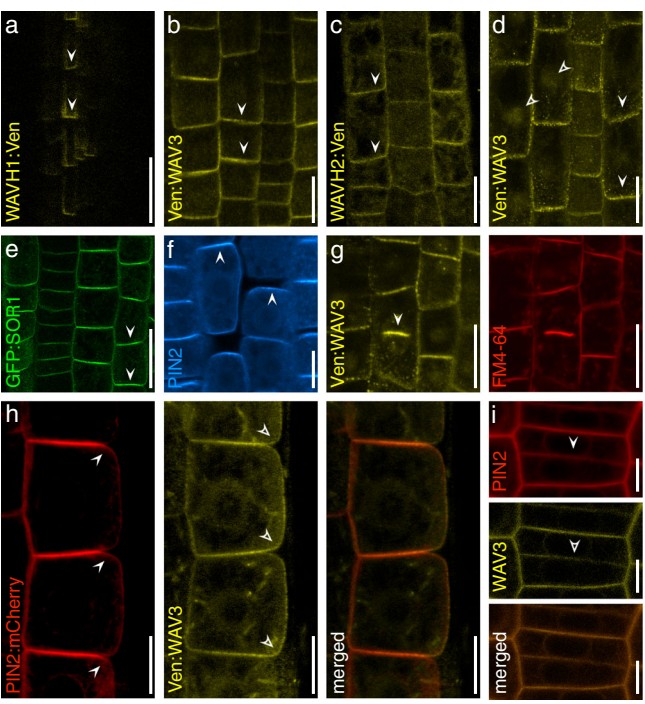

**Fig. 7 | Subcellular localization of WAV3/WAVH/SOR1 reporter proteins. a** Root meristem stele cells expressing *WAVH1::WAVH1:Ven* at 5 DAG. Root meristem epidermis cells of 5 DAG seedlings expressing *WAV3::Ven:WAV3* (**b**); *WAVH2::-WAVH2:Ven* (**c**); *XVE»Ven:WAV3* in presence of 2 μM estradiol (**d**). Closed arrowheads indicate reporter localization proximal to the PM, open arrowheads indicate intracellular reporter signals. **e** Subcellular localization of GFP:SOR1 in root meristem epidermis cells of 5 DAG *wav triple 35S::GFP:SOR1* seedlings (arrowheads). **f** PIN2 immunolocalization in root meristem epidermis cells of 5 DAG *wav triple 35S::GFP:SOR1* seedlings (arrowheads). **g** Subcellular localization of Ven:WAV3 in a dividing root meristem cell of 5 DAG *wav triple XVE»Ven:WAV3* (left, arrowhead indicates cell plate). FM4-64 co-staining was employed to highlight the emerging cell plate (right). **h** Localization of PIN2:mCherry and Ven:WAV3 in root meristem epidermis cells of 5 DAG *PIN2::PIN2:mCherry XVE»Ven:WAV3* seedlings grown in presence of 2 μM estradiol. Closed arrowheads indicate PIN2:mCherry polarity at the PM, whilst open arrowheads indicate polarity of Ven:WAV3. **i** Localization of PIN2:mCherry and Ven:WAV3 in a cytokinetic root meristem epidermis cells of 5 DAG *PIN2::PIN2:mCherry XVE»Ven:WAV3* grown in presence of 2 μM estradiol. Closed and open arrowheads indicate cell plate localization of PIN2:mCherry and Ven:WAV3, respectively. Three independent experiments were performed for (**a**–**i**) with similar results. Scale bars: **a**–**d**, **f** = 10 μm; **e** = 20 μm; **h**, **i** = 5 μm.

plate, and weak signals at the basal domain. In *wav triple*, PIN2:-Dendra we detected strong signals at the cell plate and substantially weaker signals at the basal PM domain of the basally localized daughter cell, whilst in the apical cell PIN2 apical localization is restored (Fig. 8e). This distribution resembles PIN2 in wild type cytokinetic cells, and demonstrates that BFA treatment antagonizes PIN2 missorting in cytokinetic cells. Taken together, our observations can be summarized to propose a function for WAV3, predominantly preventing PIN2 and PIN1:GFP-3 entry into an ARF GEF basal sorting and secretion pathway.

## Discussion

It is the control of polar PM distribution of PIN auxin efflux transport proteins that defines directionality of auxin flow, shaping a diversity of auxin-regulated processes[1,2,23]. However, molecular determinants described so far, exert for the most part broad effects on PIN sorting and PM distribution[38–41]. Here, we provide a characterization of WAV3/WAVH RING-finger E3 ubiquitin ligases, which, similar to PID/WAGs act, specifically, in apical-vs.-basal PIN sorting decisions. In particular, these E3s function in apical PIN sorting in cytokinetic root meristem cells,

establishing WAV3/WAVH proteins as factors in directional auxin transport.

It appears straightforward to consider participation of these E3 ligases in PID/WAG-dependent PIN sorting decisions. Whilst synergistic phenotypes of *pid/wag/wav3/wavh* mutant combinations do not support such a simple scenario, they might also arise as a consequence of WAV3/WAVH and PID/WAG activities unrelated to apical PIN sorting. Rice *SOR1*, a *WAV3* ortholog, affects stability of Aux/IAA proteins[17], whilst *PID*, apart from acting in PIN polarity control, controls overall PIN auxin transport activity[14]. Nevertheless, the intermediate phenotypes observed upon combining a *PID* gain-of-function line and *wav triple* as well as the analysis of the phosphomimic pin2[S1,2,3D]:Dendra allele provide evidence for non-epistatic interactions in PIN2 sorting. The antagonistic effects of PID/WAG-mediated PIN phosphorylation and of *wav triple*, argue for WAV3/WAVH acting interdependently with, but uncoupled from these protein kinases in the control of apical PIN sorting. This is in line with published evidence, indicating that phosphorylation by PID/WAGs per se is not sufficient for sustained apical PIN localization. Specifically, localization studies performed with antibodies recognizing PIN1 phosphorylated by PID/WAGs suggested a sustained localization of such PIN1 at basal PM domains, which clearly argues for additional activities involved in apical PIN sorting decisions[6,12]. *WAV3/WAVH* genes might represent constituents of such so far elusive PIN polarity acquisition controls.

Polarity defects in *wav triple* are highly specific, implying that common elements of the cellular sorting machinery are not affected by the mutant, but distinctively concern PIN polarity determinants. However, whilst WAV3 and SOR1, have been demonstrated to function as E3 ligases in vitro[16,17], neither inhibition of proteasome activity, nor manipulation of the PIN2 ubiquitylation status affected PIN2 polarity, suggesting that ubiquitin ligase activity of WAV3/WAVH is not directly involved. Currently, a direct WAV3-PIN2 interaction *in planta* cannot be categorically excluded, but our results are also consistent with WAV3/WAVHs acting in *trans* on the polar sorting of PIN proteins. This function might impact on the PIN2 sorting in cytokinetic cells as well as on maintenance of PIN2 apical localization after completion of cell division. The apparent crosstalk between WAV3/WAVHs and BFA-sensitive ARF-GEFs is consistent with such activities. ARF GEFs are essential for activation of ARFs in cargo trafficking[42], and characterization of ARF GEF GNOM in particular, revealed its function in maintaining basal PIN localization by means of efficient cargo (re)cycling between the basal cellular pole and intracellular sorting compartment(s)[8,10,43]. The genetic interaction between *wav triple* and *gnom*[R5] together with BFA-sensitive misrouting into basal PIN sorting pathways in *wav triple*, positions the E3s at the crossroads of PID/WAG and GNOM sorting pathways, perhaps acting on licensing factors that define apical-vs.-basal PIN sorting decisions.

Our results demonstrate that *WAV3/WAVH* genes exert unique activities in the apical targeting of PIN2, a required component of fast root gravitropism, with apically sorted PIN2 orthologs functioning in gravitropic root bending, found only in flowering plants and gymnosperms[44,45]. The innovation of apical PIN targeting in root meristem cells likely necessitated the establishment of molecular switches that antagonize PIN sequestration into apolar or basal sorting pathways[44,46]. These switches may involve the clade of WAV3 RING finger E3 ubiquitin ligases. Elucidating how these E3s impact on polar PIN sorting decisions in mechanistic terms, remains subject to future research.

## Methods

### Plant material and growth conditions

Plants were grown on PNS plant nutrient agar plates[47], supplemented with 1% (w/v) agar and 1% (w/v) sucrose, in a 16 h light/8 h dark regime at 22 °C. *wav3-1 wavh1-1 wavh2-1*[16] used for experiments is derived from 4 rounds of backcrossing into Col-0. All lines and mutant combinations

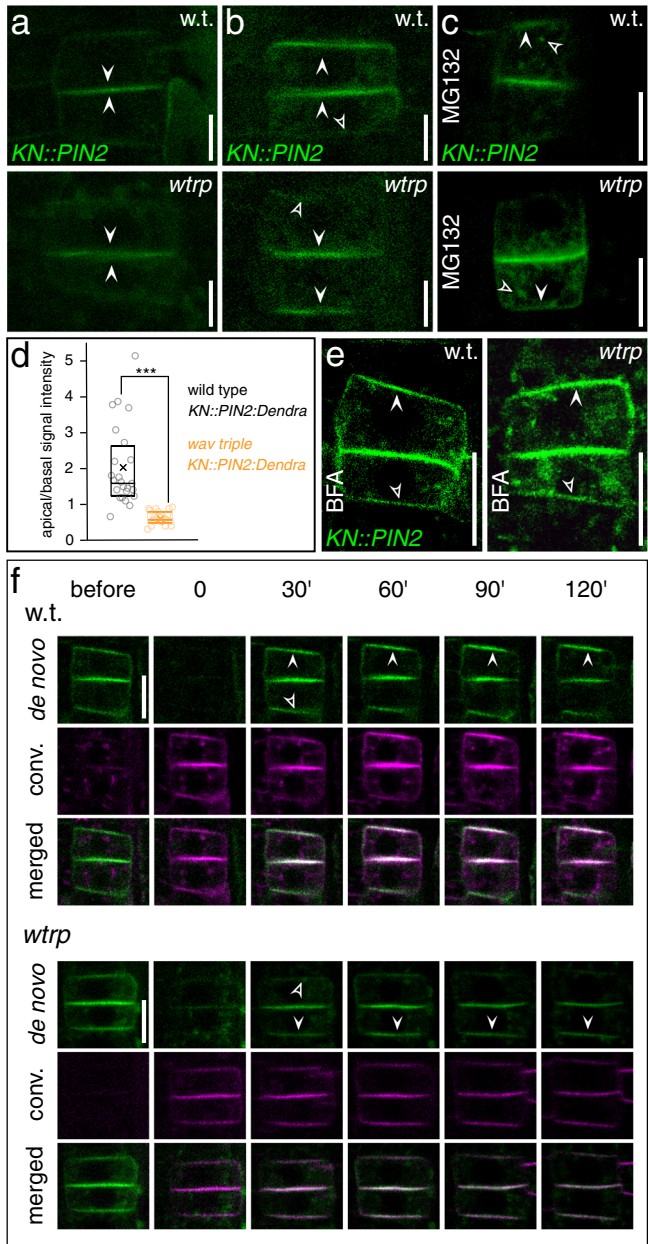

**Fig. 8 | *WAV3/WAVH* function in polar PIN2 sorting in cytokinetic root meristem cells.** PIN2:Dendra signals in dividing cells of 5 DAG *KN::PIN2:Dendra* and *wav triple KN::PIN2:Dendra*. Arrowheads indicate reporter accumulation at the cell plate (**a**) and at apical and basal domains, respectively (**b**). Open arrowheads: limited signal accumulation at the wild type basal domain and *wav triple* apical domain. **c** PIN2-Dendra signals in dividing wild type (w.t.) *KN::PIN2:Dendra* (top) and *wav triple KN::PIN2:Dendra* (bottom) root meristem epidermis cells, after treatment with 50 µM MG132 for 2 h. Open arrowheads indicate intracellular signal accumulation, closed arrowheads indicate polar localization at the PM. **d** Quantification of PIN2-Dendra polarity in wild type (*n* = 24/14 roots) and *wav triple* (*n* = 22/19 roots) dividing root meristem cells at 5 DAG; *n* = 22–24. **e** PIN2:Dendra signals in *KN::PIN2:Dendra* and *wav triple KN::PIN2:Dendra*, after treatment with 10 µM BFA for 16 h. Arrowheads indicate apical localization of reporter signals. Open arrowheads: limited signal accumulation at the wild type basal domain and *wav triple* apical domain. **f** *KN::PIN2:Dendra* photoconverted before polarity re-establishment in wild type (w.t.; top) and *wav triple* (*wtrp*; bottom), to follow the fate of de novo synthesized PIN2 for 120 min. Arrowheads indicate polar localization of de novo synthesized PIN2 at apical and basal PM domains. Open arrowheads: limited signal accumulation at the wild type basal domain and *wav triple* apical domain. 'de novo' indicates protein accumulating after conversion (green); 'conv.' represents the converted protein fraction (magenta). Open arrowheads point towards minor signal accumulation at the basal domain in wild type and the apical domain in *wav triple* cytokinetic cells (**b**, **e**, **f**). Three independent experiments were performed for (**a–c**) and (**e**, **f**) with similar results. Circles represent single data points; boxes: first and third quartiles; center line: median; 'x': mean value. Two-tailed t-test was employed to determine statistical significance (**d**); ***: *p* < 0.001. Scale bars: **a–c**, **e** = 10 µm; **f** = 5 µm. Source data are provided as Source Data file.

were confirmed by two rounds of genotyping. *DR5rev:3XVENUS-N7*[22]; *PIN2::PIN2:VEN*[30]; *PIN2::PIN2:ubq:VEN*[30]; *PIN2::pin2^K12R^:VEN*[30]; *PIN2::pin2^K12R30^*; *PIN2::PIN2:mCherry*[48]; *PIN1::PIN1:GFP*[49]; *PIN1::PIN1:GFP-2*[23]; *PIN2::PIN1:GFP-3*[23]; *BOR1::BOR1:GFP*[27]; *PEN3::PEN3:GFP*[28]; *OPS::OPS:mCit*[24]; *AUX1::YFP:AUX1*[25]; *GNOM::GNOM:GFP*[8]; *KN::PIN2:Dendra*[37]; *PIN2::PIN2:Dendra*[50]; *PIN2::pin2^S1,2,3D^:Dendra*[34], *PID::PID:YFP*[7] and *D6PKp::YFP:D6PK*[26] have been described. *eir1-4*[21], *gnom^R59^*, *wag1-1 wag2-1 pid-14*[33] have been used for crossing into *wav triple*, with segregating progeny confirmed by genotyping. Segregating *WAV3 WAVH1 WAVH2* progeny has been used as wild type control for analyses of *wav triple* phenotypes. Primers used for genotyping of plant lines are listed in Supplementary Table 2.

### Molecular cloning

Enzymes for DNA manipulation have been obtained from Thermo Fisher, Promega and NEB. For a generation of *WAV3::Venus:WAV3*, a 1400 bp *WAV3* promoter fragment plus the first 13 codons of *WAV3* was amplified from Col-0 DNA and cloned into pTZ57R/T (Fermentas). Subsequently, the Venus tag was introduced via *Spe*I after codon 13 of *WAV3* to give pTZWAV3::VENUS. In parallel, *WAV3* full-length cds was

amplified from *Arabidopsis* cDNA and cloned via *Kpn*I into the pApA binary vector[30]. The *WAV3*-promoter-Venus fragment was then introduced via *Sac*I/*Sac*II, to give pWAV3::VENUS:WAV3. For generation of *XVE»Venus:WAV3* we introduced a *Spe*I site at the 5'-end of the *WAV3* coding sequence, which was followed by introduction of the Venus tag to produce *Venus:WAV3* in pTZ57R/T. This vector served as template for Gateway recombination cloning into pDONR221 resulting in pDONR-Venus:WAV3, followed by an LR reaction into pMDC7[51] to produce pXVE»Venus:WAV3. For generation of *WAHV1::WAVH1:Venus*, we first amplified the full-length *WAVH1* coding sequence followed by ligation into pTZ57R/T and subsequent cloning into pPZP-Venus-pApA via *Xma*I. A *WAVH1* promoter fragment flanked by *Eco*RI sites was amplified from genomic DNA and combined with *WAVH1:Venus* in pPZP-Venus-pApA to give pWAHV1::WAVH1:Venus binary vector. For a generation of *WAVH2::WAVH2:Venus* a genomic *WAVH2* fragment was amplified from Col-0 DNA and cloned into pTZ57R/T. Subsequently, the *WAVH2* locus was introduced into pPZP-Venus-pApA via *Xma*I to give pWAVH2::WAVH2:Venus. AK243021, a rice *SOR1* full length cDNA clone was obtained from the Genetic Resource Center (NARO, Japan), amplified and cloned into pDONR221. Confirmed clones were then used for LR reactions and recombined with pB7WGF2[52], resulting in *35S::SOR1:GFP*. Gateway assembly was used to generate the pPIN2::X-VE»OPL2:sYFP, employing the pK8m34GW-FAST destination vector (https://gatewayvectors.vib.be/collection/pk8m34gw-fast), and as entry clones OPL2-pDONR221 (containing the *OPL2* coding sequence amplified by PCR), p1R4-ML:XVE_pPIN2 (containing a 1397 *PIN2* promoter fragment inserted into the p1R4-ML:XVE vector[53] via *Age*I/*Xho*I) and YFP-pDONRP2RP3[54]. Primers obtained for DNA cloning work are listed in Supplementary Table 2.

### Imaging and immunostaining

CLSM images were generated using Leica SP5 and SP8 (Leica Microsystems, Wetzlar, Germany) microscopes. For imaging, we used the following excitation conditions: 488 nm (GFP, mCitrine), 514 nm (Venus; YFP), 561 nm (mCherry, RFP and FM4-64).

Immunostaining has been performed with 5 day-old-seedlings[21]. After fixation in 4% (w/v) para-formaldehyde, seedlings were treated with 2% (w/v) Driselase mix (Sigma-Aldrich, D9515), followed by

solubilization in presence of 3% (v/v) Nonidet-P40 and 10% (v/v) DMSO and blocking in presence of 3% (w/v) Bovine Serum Albumin in Microtubule-stabilizing buffer (50 mM PIPES; 5 mM MgSO$_4$; 5 mM EGTA, pH 6.9). For immunostaining, we employed rabbit anti-PIN1 serum[49] (1:500) and rabbit anti-PIN2 serum[55] (1:500). As a secondary antibody we used Fluorescein isothiocyanate-(FITC)-labeled goat-anti-rabbit (1:300; Dianova, DAB-088166).

For confocal imaging and photoconversion, seedlings were placed on chambered cover glass (VWR, Kammerdeckgläser, Lab-Tek™, Nunc™ - Eine Kammer, 734-2056)[56]. With the chamber, a block of solid MS media was cut out and 10–15 seedlings were transferred to the agar, and the block was inserted into the chamber. For BFA treatment seedlings were transferred on solid MS medium containing 10 μM BFA (Sigma) and incubated for 16 h o/n. Dendra photoconversion was performed as described previously[37]. Confocal imaging was performed with Zeiss LSM800 inverted microscopes using 20× objective. Detection of fluorescence signals was carried out for green (excitation 488 nm, emission 507 nm) and red (excitation 536 nm, emission 617 nm). Pre-existing *KN::PIN2-Dendra* signal was bleached with 488 nm laser using 100% laser power and 10 iterations. After 1 h, epidermis cells with strong cytokinetic green Dendra signal were imaged and afterwards photoconverted using a fluorescent lamp with DAPI channel settings for 1 min. Images were acquired every 30 min up until 2 h after conversion. Images were analyzed using the ImageJ (NIH; http://rsb.info.nih.gov/ij) software. Mean gray value form the apical, basal and lateral membrane were obtained using the 'segmented line' tool. Polarity index values were calculated as the difference between apical and basal divided by lateral membrane intensity.

### Protein and gene expression analyses

For membrane protein extraction, root material was homogenized and resuspended in extraction buffer[55]. Samples were cleared by centrifugation, with the supernatant saved and the pellet re-extracted. Samples were again cleared, and combined supernatants were centrifuged to yield total membrane pellets. These were then subjected to Western blot analysis, transferred to nitrocellulose membranes, and probed with affinity-purified rabbit anti-PIN2[21] (1:500), followed by HRP-conjugated goat-anti-rabbit IgG (1:20,000; Jackson, 111-036-003). For comparison of protein amounts we used mouse anti-α-TUB (clone B-5-1-2; 1:1,000; Sigma, T6074), followed by HRP-conjugated goat-anti-mouse IgG (1:20,000; Jackson, 115-035-003). For analysis of PID:YFP and GNOM:GFP, protein was extracted by homogenizing 50 mg of 7 DAG seedlings in extraction buffer (50 mM Tris pH7.5, 20 mM NaCl, 0.1% Triton-X100, 10 μM PMSF, 1 × EDTA Free Roche inhibitor cocktail tablet, 1 mM DTT). The tissue was vortexed for 20 min followed by centrifugation at 14000 rpm, all at 4 °C, with the resulting supernatant used for Western blot analysis (1:1000; JL-8 anti-GFP antibody, Clontech, 632381) followed by HRP-conjugated goat-anti-mouse (1:10,000; GE healthcare, GE:NXA931).

For qPCR, RNA from 7-day-old seedlings of wild type, *wav triple*, *35S::PID* and *wav triple 35S::PID* grown on ½ MS was extracted using the SV Total RNA extraction Kit (Promega). RNA was then used for generation of cDNA by employing the iScript transcriptase kit (Bio-Rad). qRT-PCR was then performed on a qPCR qTOWER (Analytik Jena) with the *ELONGATION FACTOR 1a* (*EF1a*) as normalization control. We employed two different primer pairs for *PID* transcript level assessment and determined transcript levels for *ACT2*. Primers used for amplification are listed in Supplementary Table 2.

### Plant transformation and phenotypic analysis of lines

Flowering Arabidopsis plants were transformed with *Agrobacterium tumefaciens* (GV3101) using the floral dip method[57]. Resulting progeny was confirmed for segregating a single T-DNA insertion and cultivated to homozygosity. PCR followed by sequence analysis of relevant T-DNA fragments was employed to confirm the presence of the different *WAV3/WAVH*-derived constructs in the respective transgenic lines. For root growth assays, seedlings of each genotype were germinated on PNS in presence of the indicated compounds. For induction of *WAV3* expression, 17β-estradiol (2 μM; Sigma, E2758; unless otherwise stated) was added to the growth medium, before plating and germinating seeds. Assessment of ACC responses has been done on plates supplemented with the indicated compound concentrations. Plates were sealed with parafilm to avoid evaporation of ethylene. For comparing different genotypes, all lines tested were incubated on estradiol-supplemented medium, under identical conditions. As a solvent control, nutrient plates supplemented with DMSO have been used. After incubation on vertically positioned nutrient plates, seedlings were scanned for subsequent quantification of phenotypic traits such as root length and angles (deviation from vertical), using ImageJ software (NIH). For analyses of BFA responses, short-term treatments were made with 5-day-old seedlings incubated in liquid PNS in presence of 50 μM BFA for 90 min or o/n, after which PIN2 signals were analyzed on living material or after immunostaining. For long-term BFA treatments, 5-day-old seedlings germinated on regular medium were transferred onto fresh nutrient plates supplemented with 10 μM BFA or solvent only. Seedlings on plates were then positioned vertically in the dark. After 36 h of incubation, the direction of root growth was determined. This type of material was also used for immunostaining for analysis of subcellular PIN2 localization. For assaying BFA effects on PIN2 localization upon translational inhibition, 4-day-old *eir1-4* and *wav triple* seedlings expressing *PIN2::PIN2:VEN*, were incubated in liquid growth medium supplemented with 25 μM CHX for 60 min, followed by co-incubation in presence of 25 μM CHX and 50 μM BFA for another 90 min. As controls, we used seedlings incubated in presence of 50 μM BFA, only.

### Polar auxin transport (PAT) measurements

Root shoot-ward (basipetal) PAT measurements allowing to simultaneously quantify $^3$H-IAA and $^{14}$C-BA transport was determined using a modified protocol, initially described by Lewis and Muday[58]. Prior to the transport experiment and different from the original protocol, 7 DAG seedlings germinated on 2 μM 17β-estradiol were briefly washed in water and either depleted of 17β-estradiol by a 6 h incubation on ½ MS plates or retransferred on 17β-estradiol plates. 5% agarose beads containing each 100 nM $^3$H-IAA (specific activity 20 Ci mmol$^{-1}$; American Radiolabeled Chemicals, Inc., St. Louis, MO) and $^{14}$C-BA (specific activity of 50 Ci mmol$^{-1}$; American Radiolabeled Chemicals, Inc., St. Louis, MO), respectively, were placed in close proximity below the root tip of seedlings aligned on vertically oriented ½ MS plates supplemented with 2 μM 17β-estradiol, or on control ½ MS plates lacking estradiol and containing the solvent (DMSO) only. After 6 h, root tips were discarded and 5 mm root segments were pooled and incubated for 24 h in 5 ml scintillation cocktail and the amount of radioactivity was determined by liquid scintillation counting. Four biological replicates with each 20 seedlings per replicate were assayed on identical plates containing both wild type and mutant seedlings.

### Reproducibility and statistics

At least three biological repeats were performed for the experiments. Statistical analysis has been performed as indicated in the main text, figure legends and the Source Data file.

### Reporting summary

Further information on research design is available in the Nature Research Reporting Summary linked to this article.

## Data availability

The authors declare that the data supporting the findings of this study are available within the paper, and its Supplementary Information files. The authors declare that all data supporting the findings of this study

are available within the manuscript and its Supplementary files, and are available from the corresponding author upon request. The source data underlying Figs. 1g, j; 5f, k; 6a, d, i; 8d as well as Supplementary Figs. 1a, d, e; 2g, h; 3a, c; 4d; 5b; 6a; 7a; 8g; 9b; 11e are provided as a Source Data file. Source data are provided with this paper.

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

## Acknowledgements

We would like to thank Tatsuo Sakai, Marcus Heisler, Toru Fujiwara, Lucia Strader, Christian Hardtke, Malcolm Bennett, Claus Schwechheimer, Gerd Jürgens and Remko Offringa for sharing published materials and Alba Grau Gimeno for support. We are greatly indebted to Bert de Rybel for supporting N.K. and M.G. to work on the final stages of manuscript preparation as postdocs in his laboratory. A full-length *SOR1* cDNA clone (J090099M14) was obtained from the National Agriculture and Food Research Organization (NARO, Japan). Support by the Multiscale Imaging Core Facility at the BOKU is greatly acknowledged. This work has been supported by grants from the Austrian Science Fund (FWF P25931-B16; P31493-B25 to Christian Luschnig; I3630-B25 to Jiří Friml; P30850-B32 to Barbara Korbei) and from the Swiss National Funds (31003A-165877/1 to Markus Geisler) and the European Union's Horizon 2020 research and innovation program (Marie Skłodowska-Curie grant agreement No 885979 to Matouš Glanc).

## Author contributions

N.K., R.K., M.Glanc, S.T., K.R., J.M.A., M.S., and M.D.D. generated and analyzed constructs and plant lines. N.K., K.R., L.H., M. Glanc, and C.L. determined protein localization, abundance and sorting. B.K., M. Geisler, J.F., N.K., M. Glanc, and C.L. conceived experiments, and C.L. wrote the manuscript with input from the co-authors. All authors have had the chance to read and comment on the manuscript.

## Competing interests

The authors declare no competing interests.
