## [Peer Review File · Nature Communications]

WAVY GROWTH Arabidopsis E3 ubiquitin ligases affect apical PIN sorting decisionsREVIEWER COMMENTS

Reviewer #1 (Remarks to the Author):

In their manuscript, the authors establish a link between three E3 ubiquitin ligases WAV3/H and its putative target, a plasma membrane-localized auxin facilitator protein PIN2. They can convincingly show the importance of WAV3/H for proper apical localization of PIN2 in Arabidopsis roots which in turn ensures proper gravitropic growth responses. Also, the authors investigate the possible interdependence of polarity control by phosphorylation of PIN2 mediated by PID/WAGs and WAV3/H function on PIN2. Together with genetic experiments investigating higher-order mutants using *pid/wag* and ARF-GEF *gnom* mutants and localization studies in dividing cells using a photoconvertible PIN2 fusion protein they try to argue for placement of WAV3/H proteins at the crossroads of PID/WAG and GNOM sorting pathways.

The manuscript is well written and the experimental approaches and methodology are mostly coherent and sufficient, as are the statistics. However, some of the conclusions drawn are in my opinion too hasty. Even though the findings of this work are interesting, there would be potential for improvement in order to draw the conclusions presented by the authors.

Hence my major points as suggestions/requests for additional experimental data to substantiate the findings:

- Auxin distribution:

There are only 4 data points for each experiment, it would be important to add more measurements (the article the authors refer to uses at least 30 seedlings/data points). It would be very helpful to include a control with measurements in the *eir1-4* background. Also, in the results section (p. 5, l. 125-128) the hint on substrate-specificity for the differential distribution of only two compounds tested is too far-fetched.

- Specificity of WAV3/H PM protein sorting:

The choice of the additional polarly targeted PM reporter proteins is somewhat unfortunate as all reporters are basally localized except for OPS. The endogenous expression domain of OPS does likely not overlap with the WAV3/H expression domain. Therefore, if possible, expression of OPS or other apically expressed reporter proteins in the PIN2 expression domain as was nicely shown for PIN1 would be needed.

- The partial restoration of the *wav3/h* phenotype by PID overexpression is interesting and even though the authors go at lengths to rule out a direct connection, it would still be important to test for transcript and/or protein levels of PID in the *wav3/h* mutant background. On another note, does overexpression of WAV3 in a wt background cause PIN polarity changes?

- PIN2 polarity establishment at the end of cell division:

The pictures in Figure 5h and i are very hard to interpret and judging by the resolution it could also be seen as partial co-localization in interphase cells. Can the authors provide pictures with better resolution? Maybe it is also not of utmost importance to show colocalization since WAV3/H seemingly not directly interacts with PIN2 in a function as a ubiquitin ligase. Alternatively speaking, it is also not very surprising that proteins expressed from the KN promoter at least partially seem to colocalize at the forming cell plate.

Please also exchange the picture of the wt cell in Fig. 6e as it is not a cytokinetic cell.

Also, can the authors comment on the apical and basal signals in both the wt and *wav3/h* cytokinetic daughter cells in their photoconversion time-course experiment in Fig. 6f as the pictures in Fig. 6a-c show a different scenario?

Some minor points:

- Source data file Figure 1j: dpm instead of cpm
- p. 11 l. 326: Shouldn't it be Fig. 6c instead of d?

Reviewer #2 (Remarks to the Author):

Subcellular localization of PIN auxin transporters plays an important role in the formation of auxin distribution patterns. Although phosphorylation of PIN cytoplasmic domain by AGCVIII kinases has been shown to regulate the subcellular localization of PIN, other regulatory mechanisms are not well understood. In this study, the authors demonstrated that WAV3/WAVH E3 ligases are involved in the regulation of PIN subcellular localization in a PIN phosphorylation-independent manner. PIN2 cannot be localized at apical side of cell in wav triple mutants and the WAV3 expression compensates for the abnormal PIN2 localization in the wav3 triple mutant. Genetic and pharmacological analyses suggest that WAV3/WAVH mediate the ARF-GEF-mediated trafficking of PIN2 rather than the phosphorylation or expression of PIN2. The discovery of the regulation of WAV3/WAVH E3 ligases on the subcellular localization of PIN2 is important for understanding the overall regulation of subcellular localization of PINs. The research was well designed and carefully conducted.

Minor comments.

- 1) p4, line 89: Is a wav3-1 wavh1-1 wavh2-1 correct? (wav3-2?)
- 2) p6, line 149: Please describe what is difference between PIN1:GFP-2 and PIN1:GFP-3.
- 3) p6, line 161: Please describe what each of BOR1, PEN3, OPS, AUX1 and D6PK means.
- 4) p9, line 269: Supplementary Figure 6e and 6f do not show expression levels of GNOM in seedling. Please determine the expression levels of GNOM (or GNOM-GFP) in wild type and wav triple mutant by the immunoblotting.
- 5) p17, line 523: "(PAT) measurements" is correct.
- 6) p17, line 523: Please describe an outline on the method of PAT measurements.
- 7) Fig.3: Images of seedlings of pid-14 single mutant and wav triple mutant are necessary.
- 8) Supplementary Figure 1d: Please describe how cell division zone is determined.
- 9) Supplementary Figure 2b: These images are impressive. Those should move to Fig.2.

Reviewer #3 (Remarks to the Author):

Konstantinova and colleagues describe the striking, specific requirement of members of the Arabidopsis WAVY GROWTH3 (WAV) family of E3 ubiquitin ligases for coordinated apical PIN-FORMED2 auxin efflux facilitator sorting. The striking finding here is that PIN2 normally apically localized in root epidermal cells, elongating cortex and lateral root cap cells is almost completely misplaced to the basal membrane in a triple mutants of WAV3 and its two closest homologues. The triple mutant display root gravitropism defects which are even somewhat more pronounced than those pin2 loss-of-function mutants. The WAV proteins localize to the basal plasma membrane, where they apparently prevent PIN2 accumulation in the wild type, whereas PID/WAG-mediated phosphorylation of PIN2 promotes its recruitment to the apical domain.

Overall this is well performed work on an exciting timely research topic of broad interest. The manuscript reads very well except for a very few passages where the data may be slightly over-interpreted in light of lacking WAV overexpression results, which could be adjusted by rephrasing. The figures are very informative and well designed. In most cases they contain quantitative and statistical analysis, which should be included in a few additional places as well as some control images.

The topic of cell polarity and its coordination is not only of broad interest to the plant science community. The findings that E3 ubiquitin ligases are needed for the specific correct orientation of polarization i.e. the directionality of polarity are also likely to be of interest to cell and developmental biologists in other subject areas. Hence, the manuscript is certainly of interest to a broad readership providing new repressors of PIN2 localization at the basal domain, even if - at this point - the exact molecular mechanism by which the WAV proteins mediate this remains to be elucidated in future studies. The work thus has potential for exciting follow-up studies and is likely to be well cited.

MINOR ISSUES that do require improvement:

1) Figure 1j: The description of the auxin transport assay here is somewhat short and cryptic especially for non-specialist readers. Fig. 1j should be referred to after "estradiol" in line 119 to clarify where this can be seen and the assay (though published before) should be described briefly in the figure legend.

2) Figure 3a-e and f-i: Please, include the *pid-14*, *wtrp* and *wag1-1 wag1-2* mutants as controls here. Otherwise, it is difficult to judge for the reader, where the observed defects originate from e.g. in *pid-14 wtrp*. Also, the point of "additive effects" in line 202 is not supported if e.g. the *pid-14* single is not shown.

3) Figure 3j-r: Quantitative Data and statistical analysis such as in/from Supplementary Figure 5I should be given for ALL genotypes in Figure 3 j-r in the main text (including the numbers of roots and cells and from how many experiments), so the reader has a clear idea about the quantitative strength of the effect on apical-to-basal misplacement in the mutants or basal-to-apical replacement in e.g. the *pin2S-D-Dendra* mutants and this does not get buried in the supplementary info. I do not doubt the result, but this should be made clear here in a quantitative manner.

4) Line 170: "Our experiments identified WAV3/WAVH E3 ubiquitin ligases as unique regulators of PIN polarity". "regulators": This point cannot really be made here, as no clear-cut overexpression phenotype with respect to PIN polarity has been reported in the manuscript. So, unique "mediators" would be more appropriate.

5) line 141: "in the lower cortex cell files" should read "in the lower part of the cortex cell files"

6) line 287: "at the basal PM domain", please, clarify in the text as to whether this observation only holds true for meristematic epidermal or also cortical and lateral root cap cells. Introducing a confocal image showing this for the three proteins into the supplementary material would be helpful. Otherwise, it may take several years, such as in the case of PIN2 after its initial publication at apical membranes in epidermal and lateral root cap cells in 1998, until everybody in the field understands, where the WAVs localize in lateral root cap, epidermal, more proximal and more distal cortical cells in the meristem.

7) Abstract, line 29: "PIN-FORMED (PIN) auxin exporters". Here, "auxin efflux facilitators" would be the appropriate term as a direct exporter function has not been demonstrated, yet. ABC B type transporters may aid PIN mediated auxin efflux in heterologous systems such as yeast and *Xenopus*.

8) Abstract, line 34: "unprecedented apical-to-basal polarity switch" The authors have previously reported such a switch just not with the same completeness for *pid/wag* triple mutants. So, the priority statement should be left out and "unprecedented" may be replaced by "striking".

Reviewer #4 (Remarks to the Author):

The paper "WAVY GROWTH Arabidopsis E3 ubiquitin ligases specify apical-basal PIN sorting decisions" by Konstantinova et al. describes the observations of E3 Ligases mutants, which exhibit phenotypes similar to other mutants with defects in polar auxin transport. Upon closer inspection, the authors find mislocalization of PIN2 to the basal membranes in the epidermis and cortex cells. In the E3 Ligase (wav) triple mutant, this abnormal PIN2 localization can be reverted after BFA treatments. Similarly, when the wav triple mutant is crossed with the gnomR5 mutant, PIN2 also regains its apical localization.

Overall, this manuscript reads well and while it tells an interesting story - it is not complete.

Major issues:

-The paper fails to connect the direct action of E3 Ligases on a specific target directly affecting PIN2 polarization. The effect is evidently indirect, however it could be very interesting if the authors show the E3 Ligase activity, for example, regulating GNOM. The paper does not lack interesting observations; however, the way it is structured and conducted leads to a void of information. The main questions are not answered. Why do E3 ligases affect PIN2 localization and, as a consequence, auxin transport?

-The title is misleading; there is no specific action on PIN sorting, only an indirect action on PIN2 localization.

Perhaps a more appropriate title would be: WAVY GROWTH Arabidopsis E3 ubiquitin ligases influence apical-basal PIN2 sorting decisions.

-Why is WAV3::Ven:WAV3 only shown in wav triple mutant? What about WAV3::Ven:WAV3 in the WT background? Does it have a similar distribution/expression?

-Can the altered expression of DR5rev:3XVENUS-N7 in the distal portions of the wav triple root meristems be rescued by the XVE>>Venus:WAV3 estradiol? That result could better support the author's hypothesis.

- Line 302: In interphase cells we failed to observe overlaps in the localization of PM-associated WAV3/SOR1 reporter proteins, which are enriched at the basal PM domain, and PIN2, which for the most part locates to the apical PM domain (Fig. 5h).

We can see co-localization; however, it is hard to separate the two fluorescence signals in the middle apical/basal membrane when looking at the images displayed. This makes their argument less convincing. Some other displays or immunolocalizations could perhaps be more persuasive to the reader.

-The BFA experiments in figure 4 should be performed with PIN2:DEN to be able to distinguish if protein accumulated at the BFA bodies corresponds to newly synthesized or recycled proteins, or both. Alternatively, co-treatment with CHX could be shown. This could indicate if E3 Ligases activity is essential for secretion of newly synthesized proteins or secretion after endocytosis (recycling).

-Line 343: PIN2 accumulated at the apical PM domain in response to BFA treatment, strikingly resembling PIN2 sorting in wild type cytokinetic cells (Fig. 6e)

In the picture shown, PIN2 indeed looks apical but also is located towards the basal domain in the lower daughter cell. Does this indicate dual polarity? Should this be explored or at least acknowledged?

-Is GNOM a target of the E3 Ligases? If this is the case, it would be consistent with the BFA phenotypes and the gnomR5 wtrp phenotypes. If this question is addressed, the paper could be

substantially improved.

Minor issues:

-Graphs in the main and Supplemental Figures have an error in the display of the significance *; the stars appear to be hidden behind the lines; please correct this. Black circles representing single data points with significant letters (a, b, c) also in black are not displayed correctly and hide the letters. We suggest changing the color of the circles.

-Supplemental Figure 2A, western blot needs quantification of triplicate experiments.

-Figure 6 could likely be better seen in green and magenta (in the individual channels); we find the gray is confusing

-In line 341, change from "now longer" to "no longer"

Rebuttal Letter

Reviewer #1 (Remarks to the Author):

In their manuscript, the authors establish a link between three E3 ubiquitin ligases WAV3/H and its putative target, a plasma membrane-localized auxin facilitator protein PIN2. They can convincingly show the importance of WAV3/H for proper apical localization of PIN2 in Arabidopsis roots which in turn ensures proper gravitropic growth responses. Also, the authors investigate the possible interdependence of polarity control by phosphorylation of PIN2 mediated by PID/WAGs and WAV3/H function on PIN2. Together with genetic experiments investigating higher-order mutants using *pid/wag* and ARF-GEF *gnom* mutants and localization studies in dividing cells using a photoconvertible PIN2 fusion protein they try to argue for placement of WAV3/H proteins at the crossroads of PID/WAG and GNOM sorting pathways.

The manuscript is well written and the experimental approaches and methodology are mostly coherent and sufficient, as are the statistics. However, some of the conclusions drawn are in my opinion too hasty. Even though the findings of this work are interesting, there would be potential for improvement in order to draw the conclusions presented by the authors.

Authors' response: Thank you very much for your encouraging response and your highly elaborate and constructive review. Below, please find our responses to the individual points raised.

Hence my major points as suggestions/requests for additional experimental data to substantiate the findings:

- Auxin distribution:

There are only 4 data points for each experiment, it would be important to add more measurements (the article the authors refer to uses at least 30 seedlings/data points). It would be very helpful to include a control with measurements in the *eir1-4* background. Also, in the results section (p. 5, l. 125-128) the hint on substrate-specificity for the differential distribution of only two compounds tested is too far-fetched.

Authors' response: Thank you very much for pointing this out! We agree that our description of the PAT assays was a bit minimalistic. The revised version comes with a more accurate description of these assays, specifically in the Materials and Methods section and with more detailed Figure Legends. This includes a more detailed explanation of the setup of the BA experiments, which we have been using as an internal control for our PAT assays. Each data point shown in the graphs in Fig. 1j and Supplementary Fig. 2h, corresponds to an independent biological replicate with 20 seedlings each. As described in the original paper (Lewis and Muday, 2009), these seedlings have been pooled for scintillation counting. Different from the original protocol we used 20 instead of 30 seedlings for each of our biological repeats, as this setup allowed us to measure PAT for wild type and mutant lines on the very same plate, which we consider essential for a direct comparison of the different lines (and keeping in mind that agarose beads with the radiotracer need to be placed below each individual root tip). In total, data presented correspond to 60-80 seedlings/roots per genotype that have been processed in a highly uniform manner.

As suggested, we now have included the *eir1-4* negative control in our PAT assays (Supplementary Fig. 2h), for which we made use of exactly the same experimental conditions that have been used for wild type and *wav triple XVE>>Venus:WAV3*. These experiments demonstrated that neither estradiol treatment, nor its depletion resulted in any considerable effect on defective shootward auxin transport associated with this *pin2* null allele.

We do agree with reviewer 1: Our conclusion regarding specificity appears a bit too strong and we therefore modified the statement in our revised version.

- Specificity of WAV3/H PM protein sorting:

The choice of the additional polarly targeted PM reporter proteins is somewhat unfortunate as all reporters are basally localized except for OPS. The endogenous expression domain of OPS does likely not overlap with the WAV3/H expression domain. Therefore, if possible, expression of OPS or other apically expressed reporter proteins in the PIN2 expression domain as was nicely shown for PIN1 would be needed.

Authors' response: This is another essential point, which we addressed in the revised version. We introduced a translational reporter OPL2:YFP fusion, conditionally expressed in the *PIN2* expression domain in root meristems. Comparison of reporter signals in wild type and *wav triple* demonstrated predominantly apical localization in both genotypes. This again implies a certain specificity of WAV3/WAVH functions in the apical cargo sorting in Arabidopsis root meristems. The novel data can be found as Fig. 3f in our revised m/s.

- The partial restoration of the *wav3/h* phenotype by PID overexpression is interesting and even though the

authors go at lengths to rule out a direct connection, it would still be important to test for transcript and/or protein levels of PID in the *wav3/h* mutant background. On another note, does overexpression of WAV3 in a wt background cause PIN polarity changes?

Authors' response: Some additional experiments addressing the relationship between WAV3/WAVH and PID have been added. qPCR, testing *PID* transcript levels yielded no striking difference between *35S::PID* and *wav triple 35S::PID* lines - Supplementary Fig. 5b in our revised manuscript. We also added triplicate Western blots with protein extracts from wild type and *wav triple* expressing a *PID::YFP:PID* reporter construct. No differences in the abundance of YFP:PID could be detected in these 3 biological repeats, indicating that WAV3/WAVH E3 ligases do not impact on overall PID protein levels. Furthermore, when viewing sub-cellular distribution of YFP:PID, we did not detect any obvious alterations in *wav triple*, arguing for limited effects of WAV3/WAVH on the protein kinase. Results from these experiments are summarized in Supplementary Fig. 6a-c. We also did qPCR on the *35S::PID* lines. No striking differences in *PID* transcript levels in *35S::PID* and *wav triple 35S::PID* seedlings were observed (Supplementary Fig. 5b).

We also addressed consequences of WAV3 overexpression on PIN2 localization in wild type. Estradiol-inducible *Ven:WAV3* in wild type exhibits a subcellular distribution that is indistinguishable from the situation in *wav triple*. Furthermore, when doing PIN2 immunostainings on wild type (estradiol-induced) *XVE>>Venus:WAV3* roots, we observed unaffected subcellular PIN2 localization, indicating that expression of an extra copy of WAV3 in wild type does not cause a dosage-dependent apicalization of PIN2 subcellular localization. Results from these experiments are shown in Supplementary Figure 10a-e.

- PIN2 polarity establishment at the end of cell division:

The pictures in Figure 5h and i are very hard to interpret and judging by the resolution it could also be seen as partial co-localization in interphase cells. Can the authors provide pictures with better resolution? Maybe it is also not of utmost importance to show colocalization since WAV3/H seemingly not directly interacts with PIN2 in a function as a ubiquitin ligase. Alternatively speaking, it is also not very surprising that proteins expressed from the KN promoter at least partially seem to colocalize at the forming cell plate.

Please also exchange the picture of the wt cell in Fig. 6e as it is not a cytokinetic cell.

Also, can the authors comment on the apical and basal signals in both the wt and *wav3/h* cytokinetic daughter cells in their photoconversion time-course experiment in Fig. 6f as the pictures in Fig. 6a-c show a different scenario?

Authors' response: Thank you very much for pointing this out. Indeed, we have tried to establish evidence for a direct interaction between WAV3 and PIN2, but so far there are no such indications, which we also mention in the discussion section of the current manuscript. Venus:WAV3 signals at the cell plate might point towards a function already at early stages of PIN2 polarity establishment. This, of course does not necessarily require a direct interaction between PIN2 and WAV3. We rephrased this section in the revised manuscript, as there is currently no evidence for a direct interaction between these proteins.

We have replaced former Fig. 5h/i panels by pictures, where the differences in PIN2:mCherry and Venus:WAV3 localization is more obvious (Fig. 7h,i in the revised manuscript). Evidently, we cannot entirely exclude co-localization/interaction of Venus:WAV3 and PIN2:mCherry. However, Venus:WAV3 signals at the basal plasma membrane domain vs. apical localization of PIN2:mCherry in somatic root meristem epidermis cells is more evident in those new images. We also replaced Fig. 6e -now Fig. 8e- in the revised manuscript.

The KN::PIN2:Dendra photoconversion experiments on display were among the technically most demanding that we present in our manuscript. We noted a limited variability in those experiments, which might reflect certain variations in amounts of converted protein already at the plasma membrane or still *en route* in sorting or recycling compartments. Variations in PIN2 deposition in our images (comparing Fig. 8b,c and Fig. 8 e,f) might arise as a result of differing experimental set-ups being used (see Methods section). In fact, apical vs. basal sorting of *de novo* synthesized reporter proteins does not appear to be mutually exclusive, indicated by weak signals at the 'wrong' PM domains visible in Fig. 8f. Similarly, when viewing cytokinetic wild type or *wav triple* cells (Fig. 8b), one can observe faint signals at the 'wrong' plasma membrane domain (basal in wild type and apical in *wav triple*), though at a much weaker intensity. Mechanisms and significance underlying such sorting pathways remain to be elucidated. The main and very consistent point that we wanted to stress from these experiments, is that *de novo* synthesized PIN2 in *wav triple* cytokinetic cells undergoes a different fate than in wild type cells, reflected in a very strong tendency to get sorted to the basal pole of such cells. We have modified this section accordingly.

Some minor points:

- Source data file Figure 1j: dpm instead of cpm

Authors' response: Has been corrected.

- p. 11 l. 326: Shouldn't it be Fig. 6c instead of d?

Authors' response: Has been modified accordingly in the revised manuscript.

Reviewer #2 (Remarks to the Author):

Subcellular localization of PIN auxin transporters plays an important role in the formation of auxin distribution patterns. Although phosphorylation of PIN cytoplasmic domain by AGCVIII kinases has been shown to regulate the subcellular localization of PIN, other regulatory mechanisms are not well understood. In this study, the authors demonstrated that WAV3/WAVH E3 ligases are involved in the regulation of PIN subcellular localization in a PIN phosphorylation-independent manner. PIN2 cannot be localized at apical side of cell in *wav* triple mutants and the WAV3 expression compensates for the abnormal PIN2 localization in the *wav3* triple mutant. Genetic and pharmacological analyses suggest that WAV3/WAVH mediate the ARF-GEF-mediated trafficking of PIN2 rather than the phosphorylation or expression of PIN2. The discovery of the regulation of WAV3/WAVH E3 ligases on the subcellular localization of PIN2 is important for understanding the overall regulation of subcellular localization of PINs. The research was well designed and carefully conducted.

Authors' response: Thank you very much indeed for this very supportive feedback! Please, find our responses to your comments below.

Minor comments.

1) p4, line 89: Is a *wav3-1 wavh1-1 wavh2-1* correct? (*wav3-2?*)

Authors' response: Thank you for pointing this out. *Wav3-1 wavh1-1 wavh2-1* is correct. After repeated backcrossing into Col-0, we have been using this line for our analyses.

2) p6, line 149: Please describe what is difference between PIN1:GFP-2 and PIN1:GFP-3.

Authors' response: In the revised manuscript, we now mention the differences in the positioning of the GFP tag in these two constructs.

3) p6, line 161: Please describe what each of BOR1, PEN3, OPS, AUX1 and D6PK means.

Authors' response: We have added the requested information, with the subcellular positioning of the different reporter proteins now summarized in the main text.

4) p9, line 269: Supplementary Figure 6e and 6f do not show expression levels of GNOM in seedling. Please determine the expression levels of GNOM (or GNOM-GFP) in wild type and *wav* triple mutant by the immunoblotting.

Authors' response: We apologize for not having included this important information in our original manuscript! Supplementary Fig. 8g, now shows our Western blot analysis of *wav triple* and corresponding wild type expressing GNOM::GNOM:GFP. No differences in the relative abundance of GNOM:GFP could be detected in three biological repeats of this experiment.

5) p17, line 523: "(PAT) measurements" is correct.

Authors' response: Thank you very much for noticing! We now have added the missing bracket!

6) p17, line 523: Please describe an outline on the method of PAT measurements.

Authors' response: As requested also by Reviewer 1 and 3, we have revised the section describing PAT measurements in root meristems. The experimental setup is based on protocols first published by the Muday lab (Lewis and Muday 2009), and our revised version now provides a detailed description in the Methods section, and essential information is also provided in the legends of Fig. 1 and of Supplementary Fig. 2.

7) Fig.3: Images of seedlings of *pid-14* single mutant and *wav* triple mutant are necessary.

Authors' response: In the revised version, we have added images of the *pid-14* single mutant and of the *wav triple* mutant. These images are now part of our newly generated Fig. 4a-h.

8) Supplementary Figure 1d: Please describe how cell division zone is determined.

Authors' response: Thank you very much for pointing this out! We did not differentiate between root meristem cell division and transition zone, but rather defined the onset of cell elongation, i.e. the area in which epidermis cells no longer exhibit almost isodiametric shapes. We have modified Legend and Figure accordingly.

9) Supplementary Figure 2b: These images are impressive. Those should move to Fig.2.

Authors' response: We are happy to follow this suggestion! Former Supplementary Fig. 2 is now Fig. 3 in our revised manuscript. Furthermore, we have added a localization analysis of OPL2::sYFP conditionally expressed in the *PIN2* expression domain. No differences in the (apical) localization of the reporter protein in wild type and *wav triple* root meristem epidermis cells could be observed (Fig. 3f), further demonstrating a role for *WAV3/WAVH* in apical targeting of only a subset of polarly localized membrane proteins.

Reviewer #3 (Remarks to the Author):

Konstantinova and colleagues describe the striking, specific requirement of members of the Arabidopsis WAVY GROWTH3 (WAV) family of E3 ubiquitin ligases for coordinated apical PIN-FORMED2 auxin efflux facilitator sorting. The striking finding here is that PIN2 normally apically localized in root epidermal cells, elongating cortex and lateral root cap cells is almost completely misplaced to the basal membrane in a triple mutants of *WAV3* and its two closest homologues. The triple mutant display root gravitropism defects which are even somewhat more pronounced than those *pin2* loss-of-function mutants. The WAV proteins localize to the basal plasma membrane, where they apparently prevent PIN2 accumulation in the wild type, whereas PID/WAG-mediated phosphorylation of PIN2 promotes its recruitment to the apical domain.

Overall this is well performed work on an exciting timely research topic of broad interest. The manuscript reads very well except for a very few passages where the data may be slightly over-interpreted in light of lacking WAV overexpression results, which could be adjusted by rephrasing. The figures are very informative and well designed. In most cases they contain quantitative and statistical analysis, which should be included in a few additional places as well as some control images.

The topic of cell polarity and its coordination is not only of broad interest to the plant science community. The findings that E3 ubiquitin ligases are needed for the specific correct orientation of polarization i.e. the directionality of polarity are also likely to be of interest to cell and developmental biologists in other subject areas. Hence, the manuscript is certainly of interest to a broad readership providing new repressors of PIN2 localization at the basal domain, even if - at this point - the exact molecular mechanism by which the WAV proteins mediate this remains to be elucidated in future studies. The work thus has potential for exciting follow-up studies and is likely to be well cited.

Authors' response: Thank you very much indeed for this positive feedback and encouragement! We have tried to address all the points raised by reviewer #3 in our revised m/s. Our point-by-point response can be found below.

MINOR ISSUES that do require improvement:

1) Figure 1j: The description of the auxin transport assay here is somewhat short and cryptic especially for non-specialist readers. Fig. 1j should be referred to after "estradiol" in line 119 to clarify where this can be seen and the assay (though published before) should be described briefly in the figure legend.

Authors' response: As requested also by Reviewer 1 and 2, we have revised the section describing PAT measurements in root meristems. The experimental setup is based on protocols first published by the Muday lab (Lewis and Muday 2009), and our revised version now provides a detailed description in the Methods section, and essential information is also provided in the legends of Fig. 1 and of Supplementary Fig. 2.

2) Figure 3a-e and f-i: Please, include the *pid-14*, *wtrp* and *wag1-1 wag1-2* mutants as controls here. Otherwise, it

is difficult to judge for the reader, where the observed defects originate from e.g. in *pid-14 wtrp*. Also, the point of “additive effects” in line 202 is not supported if e.g. the *pid-14* single is not shown.

Authors' response: Thank you for pointing this out! In the original version of our m/s we had seedlings of those genotypes in the Supplementary datasets only. The revised version now has these mutants on display in the main manuscript (as part of revised Fig. 4a-h).

3) Figure 3j-r: Quantitative Data and statistical analysis such as in/from Supplementary Figure 5l should be given for ALL genotypes in Figure 3 j-r in the main text (including the numbers of roots and cells and from how many experiments), so the reader has a clear idea about the quantitative strength of the effect on apical-to-basal misplacement in the mutants or basal-to-apical replacement in e.g. the *pin2S-D-Dendra* mutants and this does not get buried in the supplementary info. I do not doubt the result, but this should be made clear here in a quantitative manner.

Authors' response: This is another point that we missed in the original version of our manuscript. In the revised version, we have added the required quantification for both, *35S::PID* in *wav triple* and for the *PIN2::Dendra* alleles expressed in *eir1-4* and *eir1-4 wav triple*. These analyses have been included in our revised manuscript as Fig. 5f & k.

4) Line 170: “Our experiments identified WAV3/WAVH E3 ubiquitin ligases as unique regulators of PIN polarity”. “regulators”: This point cannot really be made here, as no clear-cut overexpression phenotype with respect to PIN polarity has been reported in the manuscript. So, unique “mediators” would be more appropriate.

Authors' response: Agreed! We replaced the term 'regulators', which now reads 'mediators', instead. In the revised m/s, we have added some analysis of estradiol-inducible Venus:WAV3 expressed in wild type. No striking effects on PIN2 polarity could be observed in these experiments (Supplementary Fig. 10).

5) line 141: “in the lower cortex cell files” should read “in the lower part of the cortex cell files”

Authors' response: We have edited this section accordingly in our revised m/s.

6) line 287: “at the basal PM domain”, please, clarify in the text as to whether this observation only holds true for meristematic epidermal or also cortical and lateral root cap cells. Introducing a confocal image showing this for the three proteins into the supplementary material would be helpful. Otherwise, it may take several years, such as in the case of PIN2 after its initial publication at apical membranes in epidermal and lateral root cap cells in 1998, until everybody in the field understands, where the WAVs localize in lateral root cap, epidermal, more proximal and more distal cortical cells in the meristem.

Authors' response: Thank you for pointing this out! In the revised version of our m/s we had another image of a *wav triple WAV3::Venus:WAV3* root meristem. Apart from cytoplasmic signals, signals appear prominent at the plasma membrane, with signals located predominantly to the basal cellular domain in all cells showing expression of the reporter protein. This image can be found as Supplementary Fig. 9a in our revised manuscript.

7) Abstract, line 29: “PIN-FORMED (PIN) auxin exporters”. Here, “auxin efflux facilitators” would be the appropriate term as a direct exporter function has not been demonstrated, yet. ABC B type transporters may aid PIN mediated auxin efflux in heterologous systems such as yeast and *Xenopus*.

Authors' response: We have edited this section accordingly in our revised m/s.

8) Abstract, line 34: “unprecedented apical-to-basal polarity switch” The authors have previously reported such a switch just not with the same completeness for *pid/wag triple* mutants. So, the priority statement should be left out and “unprecedented” may be replaced by “striking”.

Authors' response: We have edited this section accordingly in our revised m/s.

Reviewer #4 (Remarks to the Author):

The paper "WAVY GROWTH Arabidopsis E3 ubiquitin ligases specify apical-basal PIN sorting decisions" by

Konstantinova et al. describes the observations of E3 Ligases mutants, which exhibit phenotypes similar to other mutants with defects in polar auxin transport. Upon closer inspection, the authors find mislocalization of PIN2 to the basal membranes in the epidermis and cortex cells. In the E3 Ligase (*wav*) triple mutant, this abnormal PIN2 localization can be reverted after BFA treatments. Similarly, when the *wav* triple mutant is crossed with the *gnomR5* mutant, PIN2 also regains its apical localization.

Overall, this manuscript reads well and while it tells an interesting story - it is not complete.

Authors' response: We are happy that our m/s left an overall positive impression, and we tried to address all the constructive input that has been brought up by Reviewer #4. Please find our point-by-point response below.

Major issues:

-The paper fails to connect the direct action of E3 Ligases on a specific target directly affecting PIN2 polarization. The effect is evidently indirect, however it could be very interesting if the authors show the E3 Ligase activity, for example, regulating GNOM. The paper does not lack interesting observations; however, the way it is structured and conducted leads to a void of information. The main questions are not answered. Why do E3 ligases affect PIN2 localization and, as a consequence, auxin transport?

Authors' response: We entirely agree with the reviewer! For more than four years now, we have tried to establish this missing link between PIN2 polarity control and the function of the WAV3/WAVH E3 ligases. This involved approaches in which we tested for a direct interaction between PIN2 and WAV3, for which we have not obtained experimental evidence this far, as well as addressing crosstalk with 'obvious' targets for WAV3 E3 ligase activity in the control of PIN2 polarity.

In this revised version we now have added additional experiments in which we asked if loss of *WAV3/WAVH* genes could affect the localization and/or abundance of protein kinase PINOID (as it functions as a regulator of apical PIN sorting) as well as of ARF GEF GNOM (as its function seems to be essential for basal sorting of PIN2 in *wav triple*). Westerns in which we determined abundance of YFP-PID and GNOM-GFP protein levels in wild type and *wav triple* in three biological repeats did not reveal any differences in the accumulation of these reporter proteins (as shown in Supplementary Figs. 6a and 8g). Similarly, we did not observe any striking differences in the subcellular distribution of these reporter proteins, when expressed in wild type or *wav triple* (as shown in Supplementary Figs. 6b,c and 8e,f). These results do not support simple models, in which modulation of PID or GNOM by activity of WAV3/WAVH E3 ligases would impact on the polarity of apical PIN2.

We therefore have initiated additional approaches in which we try to reveal mechanisms by which WAV3/WAVH might impact on PIN2 polarity. These involve yeast split-ubiquitin screens, employing WAV3 as a bait, as well as Venus:WAV3 IP from membrane and soluble root protein extracts, followed by identification of co-precipitated proteins by MS. The outcome of these experiments led to some potentially promising links between WAV3 and the sorting machinery in Arabidopsis. However, demonstrating such regulatory cross-talk and testing its functional significance for PIN2 polarity control, appears beyond the scope of the current manuscript.

-The title is misleading; there is no specific action on PIN sorting, only an indirect action on PIN2 localization. Perhaps a more appropriate title would be: WAVY GROWTH Arabidopsis E3 ubiquitin ligases influence apical-basal PIN2 sorting decisions.

Authors' response: We have modified the misleading wording in the title in our revised manuscript.

-Why is WAV3::Ven:WAV3 only shown in *wav triple* mutant? What about WAV3::Ven:WAV3 in the WT background? Does it have a similar distribution/expression?

Authors' response: Thank you for pointing this out! In our revised manuscript we have added information on expression of *WAV3::Ven:WAV3* and *XVE>>Ven:WAV3* in wild type. Both reporter proteins exhibit a subcellular localization as described for *wav triple*. Furthermore, when determining PIN2 localization in wild type *XVE>>Ven:WAV3* we did not observe any obvious differences when comparing induced (estradiol) and control settings (DMSO). These new findings are presented as Supplementary Fig 10a-e in our revised manuscript.

-Can the altered expression of *DR5rev:3XVENUS-N7* in the distal portions of the *wav triple* root meristems be rescued by the *XVE>>Venus:WAV3* estradiol? That result could better support the author's hypothesis.

Authors' response: We agree! This represents a valuable control for testing a role of WAV3 in PAT. We introduced *XVE>>Venus:WAV3* into *wav triple DR5rev:3XVENUS-N7* and analyzed reporter signals upon induction of Ven:WAV3. Similar to the results obtained in our comparison of wild type and *wav triple* root meristems, we detected ectopic DR5-Venus signals in lateral portions of *wav triple XVE>>Venus:WAV3 DR5rev:3XVENUS-N7*, when incubated in presence of DMSO. Induction by estradiol, resulted in a disappearance of such ectopic signals, consistent with a scenario in which establishment of correct PIN2 localization causes a

restoration of shootward PAT in root meristems. Results from these experiments have been summarized in Supplementary Fig. 2a-d.

- Line 302: In interphase cells we failed to observe overlaps in the localization of PM-associated WAV3/SOR1 reporter proteins, which are enriched at the basal PM domain, and PIN2, which for the most part locates to the apical PM domain (Fig. 5h).

We can see co-localization; however, it is hard to separate the two fluorescence signals in the middle apical/basal membrane when looking at the images displayed. This makes their argument less convincing. Some other displays or immunolocalizations could perhaps be more persuasive to the reader.

Authors' response: In our revised manuscript, we have replaced those images. Images on display in Fig. 7h (former Fig. 5) exhibit an improved clarity, showing opposing polar localization of PIN2:mCherry and Ven:WAV3 signals in root meristem epidermis cells. Furthermore, new images showing PIN2 and WAV3 reporter localization in proximity of the cell plate of cytokinetic cells have been used for our revised manuscript (Fig. 7i). Of course we cannot categorically exclude PIN2-WAV3 co-localization, and it was not our intention to leave this particular impression. We therefore carefully revised those sections in our manuscript.

-The BFA experiments in figure 4 should be performed with PIN2:DEN to be able to distinguish if protein accumulated at the BFA bodies corresponds to newly synthesized or recycled proteins, or both. Alternatively, co-treatment with CHX could be shown. This could indicate if E3 Ligases activity is essential for secretion of newly synthesized proteins or secretion after endocytosis (recycling).

Authors' response: Thank you for this suggestion! We did experiments, in which we determined BFA responses upon translational inhibition by pre/co-treatment in presence of CHX. In the control setting, i.e. PIN2-Venus in *eir1-4* we did not observe a significant difference in plasma membrane-to-BFA compartment ratios, also reflecting published observations in which apically sorted PIN2 exhibited limited responsiveness to inhibition of BFA-sensitive ARF GEFs. When using *wav triple PIN2::PIN2:VEN* seedlings for such experiments, we observed a significant decrease in the plasma membrane-to-BFA compartment ratio. This would imply that intracellular PIN2 accumulation in BFA compartments via endocytic sorting from the plasma membrane is still operational in *wav triple*, and might suggest a function for the E3 ligases in the correct sorting of newly synthesized PIN2 to the plasma membrane. These findings are signified in the main text and summarized in Supplementary Fig. 7a-e in our revised manuscript.

-Line 343: PIN2 accumulated at the apical PM domain in response to BFA treatment, strikingly resembling PIN2 sorting in wild type cytokinetic cells (Fig. 6e)

In the picture shown, PIN2 indeed looks apical but also is located towards the basal domain in the lower daughter cell. Does this indicate dual polarity? Should this be explored or at least acknowledged?

Authors' response: We also noted that there is a limited basal signal in the lower daughter cells both in BFA-treated wild type and in *wav triple* (Fig. 8e in our revised manuscript). We found a similar situation in our PIN2-Dendra conversion experiments, with limited amounts of signals at the basal domain of cytokinetic wild type cells and some very weak signals at the apical pole of the designated upper daughter cell in *wav triple* (see *de novo* synthesized protein in Fig 8f). Such signal distribution became reproducibly detectable when imaging with higher laser intensities (as used for the conversion experiments), and might indeed indicate a certain leakiness in unipolar PIN2 sorting in cytokinetic cells, which also can be seen in the manuscript by Glanc and colleagues, describing the sorting of PIN2 in cytokinetic cells. (*Nat. Plants*, 2018, 4; 1082–1088; doi.org/10.1038/s41477-018-0318-3). Specifically during earlier stages most of the PIN2 signal is found at the cell plate, with very weak signals detectable at apical and basal pole of the dividing cell pair (Glanc et al., 2018; Fig. 1e '1 hour' and '2 hours' timepoints). In addition, cellular stress induced by the harsh experimental conditions used for the experiments (inhibitor treatments and conversion, summarized in Fig. 8) could have impacted on intracellular sorting control. Further experimentation, which appears beyond the scope of this current manuscript is required to address these potentially relevant observations. We followed the reviewer's suggestions and carefully modified the wording of this section in our revised manuscript.

-Is GNOM a target of the E3 Ligases? If this is the case, it would be consistent with the BFA phenotypes and the *gnomR5 wtrp* phenotypes. If this question is addressed, the paper could be substantially improved.

Authors' response: We completely agree with the reviewer! A demonstration of altered GNOM fate in *wav triple* would link WAV3/WAVH function to the regulation of basal PIN cargo sorting. As suggested by the reviewer, we added an analysis of GNOM steady-state protein levels in *wav triple*, employing the *GNOM::GNOM:GFP* reporter

line. Consistent with an indistinguishable distribution of GNOM-GFP reporter signals in wild type and *wav triple* root meristem cells, we did not observe any differences in GNOM-GFP abundance in three biological repeats (Supplementary Fig. 8g in our revised manuscript), indicating that WAV3/WAVH E3 ligase activity does not directly impact on GNOM turnover.

Minor issues:s

-Graphs in the main and Supplemental Figures have an error in the display of the significance *; the stars appear to be hidden behind the lines; please correct this. Black circles representing single data points with significant letters (a, b, c) also in black are not displayed correctly and hide the letters. We suggest changing the color of the circles.

Authors' response: Thank you for pointing this out! We have modified all of our graphs accordingly

-Supplemental Figure 2A, western blot needs quantification of triplicate experiments.

Authors' response: Thanks for this! The revised version now includes data based on the output of triplicate Westerns for PIN2, GNOM-GFP and YFP-PID expressed either in wild type or in *wav triple*.

-Figure 6 could likely be better seen in green and magenta (in the individual channels); we find the gray is confusing

Authors' response: We have changed the coloration in our revised manuscript.

-In line 341, change from "now longer" to "no longer"

Authors' response: This section has been edited.

REVIEWERS' COMMENTS

Reviewer #1 (Remarks to the Author):

Dear authors,

Thank you for your splendid work. I am impressed by the effort in performing all additional experiments requested by myself and my fellow three reviewers. In light of thoroughly addressing everything asked for and thereby in my opinion strengthening the major findings of this work, I strongly recommend publication of this manuscript in Nature Communications.

Reviewer #2 (Remarks to the Author):

The authors have appropriately addressed all my comments and revised the paper.

Reviewer #3 (Remarks to the Author):

In this revised version of the manuscript the authors have addressed all my previous concerns concerning experimental controls, statistical analyses and textual changes. During the revision process, an error was introduced in the new text that should be taken care of prior to submission of a final version for publication (as outlined below).

I feel the authors have also satisfactorily addressed most (if not all) of the other reviewers' comments.

This is a timely, high quality manuscript on a highly topic subject area that should be well cited by plant biologists and researchers working on cell polarity and/or E3 ligase function in other organisms.

MINOR ISSUES:

1) Introduction of the genotypes in panels Figure 1h,i would facilitate understanding of the figures without the necessity of reading the legend.

2) There is an error in line 174: basal (AUXIN RESISTANT 1, AUX1::YFP:AUX1; D6 PROTEIN KINASE, D6PK::YFP:D6PK)

AUX1::YFP:AUX1 has NOT been shown to be located preferentially at BASAL domains in root cells: It is located at APICAL domains in protophloem cells and it is located ubiquitously at the plasma membrane of epidermal (mostly atrichoblast) and lateral root cap cells as well as some columella root cap cells (Swarup et al., 2004; Kleine-Vehn et al., 2006; Ikeda et al. 2009 etc.) potentially enriched slightly at apical and basal membranes of epidermal atrichoblast cells (Kleine-Vehn et al., 2006; Ikeda et al. 2009). In contrast to YFP-D6PK it should thus not be mentioned as a basal marker but rather apical or non-polar depending on the cell type.

Reviewer #4 (Remarks to the Author):

The authors have addressed all the experimental concerns from our first review. However, major issues remain regarding several unsupported claims in the text, see four examples below:

-The first is in the title: (line 1):

"WAVY GROWTH Arabidopsis E3 ubiquitin ligases affect apical-basal PIN sorting decisions"

-Also in the abstract (line 41):

"Our findings reveal so far unknown principles of PIN polarity acquisition and establish a distinct function for E3 ligases, acting in the selective targeting of polarly localized plasma membrane protein cargo in higher plants."

- In the introduction (line 76):

"Genetic and cell biological analysis reveals so far unprecedented functions for E3 ubiquitin ligases in the control of polar PM cargo sorting in general, and directional organ growth by defining PIN PM distribution in particular."

Finally, in the discussion (line 407):

"In particular, these E3s function in apical PIN sorting in cytokinetic root meristem cells, establishing WAV3/WAVH proteins as key determinants for directional auxin transport."

The authors have not shown sufficient evidence to support these claims. There is no evidence of E3 ubiquitin ligase activity directly controlling PIN polar cargo sorting. There are only indirect observations, as the mutant phenotypes could simply be attributed to an indirect result of losing E3 ubiquitin ligase activity. Additionally, there is no evidence of a general problem with sorting of all the PIN family members or all apical cargo, only with PIN2 and the ectopically expressed PIN1-GFP3. No other apical or basal sorted protein appears to be affected. This is not a minor phrasing issue, the authors need to soften their claims throughout the manuscript, or show evidence supporting these central claims. Taking into account the author's response pointing that after years of research they have not been able to produce such direct evidence, we suggest the following changes to soften the claims in the text:

-The authors altered the previous title, but the new title still needs revision for accuracy, the title could be changed to:

"WAVY GROWTH Arabidopsis E3 ubiquitin ligases activity affects apical PIN sorting decisions."

-Suggested changes to soften the claims in the abstract (line 41):

"Our findings reveal the involvement of E3 ligases in the selective targeting of apically localized PINs in higher plants."

-Suggested changes to soften the claims at the end of the introduction (line 76):

"Genetic and cell biological analyses reveals a connection between E3 ubiquitin ligase function and the control of apical PIN cargo sorting, influencing directional auxin flow and root growth."

-Suggested changes for discussion (line 407):

"In particular, these E3s function in apical PIN sorting in cytokinetic root meristem cells, establishing WAV3/WAVH proteins as factors in directional auxin transport."

Other issues that we found in the conclusions:

-In line 396 reads:

"Taken together, our observations can be summarized to propose a function for WAV3 in preventing PIN from entering BFA-sensitive ARF GEF basal sorting/recycling pathways, thereby enabling its correct positioning at apical PM domains."

Most of the evidence shown, however, indicates that the WAV3 function somehow prevents PIN from entering BFA-sensitive ARF-GEF secretory pathway. Similar to the authors' conclusion in line 292:

"Collectively, these results indicate that WAV3/WAVH function predominantly in the sorting of newly

synthesized PIN2 into the apical secretory pathway."

-we suggest changing to:

"Taken together, our results propose a function for WAV3 predominantly preventing PIN2 and PIN1-GFP3 entry into an ARF GEF basal sorting and secretion pathway".

Dear Editors,

Below please find our response to the reviewers' comments/requests (highlighted in red) for our manuscript **NCOMMS-21-50656B**.

Best regards,

Christian Luschnig, on behalf of all co-authors.

REVIEWERS' COMMENTS

Reviewer #1 (Remarks to the Author):

Dear authors,

Thank you for your splendid work. I am impressed by the effort in performing all additional experiments requested by myself and my fellow three reviewers. In light of thoroughly addressing everything asked for and thereby in my opinion strengthening the major findings of this work, I strongly recommend publication of this manuscript in Nature Communications.

Response: Thank you very much for your time and efforts!

Reviewer #2 (Remarks to the Author):

The authors have appropriately addressed all my comments and revised the paper.

Response: Thank you very much! Your feedback is greatly appreciated!

Reviewer #3 (Remarks to the Author):

In this revised version of the manuscript the authors have addressed all my previous concerns concerning experimental controls, statistical analyses and textual changes. During the revision process, an error was introduced in the new text that should be taken care of prior to submission of a final version for publication (as outlined below).

I feel the authors have also satisfactorily addressed most (if not all) of the other reviewers' comments.

This is a timely, high quality manuscript on a highly topic subject area that should be well cited by plant biologists and researchers working on cell polarity and/or E3 ligase function in other organisms.

Response: Thank you very much for this very positive feedback! We have addressed all the remaining points (see below).

MINOR ISSUES:

1) Introduction of the genotypes in panels Figure 1h,i would facilitate understanding of the figures without the necessity of reading the legend.

Response: We have added this information in the revised version of Figure 1.

2) There is an error in line 174: basal (AUXIN RESISTANT 1, AUX1::YFP:AUX1; D6 PROTEIN KINASE, D6PK::YFP:D6PK)

AUX1::YFP:AUX1 has NOT been shown to be located preferentially at BASAL domains in root cells: It is located at APICAL domains in protophloem cells and it is located ubiquitously at the plasma membrane of epidermal (mostly atrichoblast) and lateral root cap cells as well as some columella root cap cells (Swarup et al., 2004; Kleine-Vehn et al., 2006; Ikeda et al. 2009 etc.) potentially enriched slightly at apical and basal membranes of epidermal atrichoblast cells (Kleine-Vehn et al., 2006; Ikeda et al. 2009). In contrast to YFP-D6PK it should thus not be mentioned as a basal marker but rather apical or non-polar depending on the cell type.

Response: Thank you for pointing this out! We now have modified this section in our revised manuscript.

Reviewer #4 (Remarks to the Author):

The authors have addressed all the experimental concerns from our first review. However, major issues remain regarding several unsupported claims in the text, see four examples below:

-The first is in the title: (line 1):

"WAVY GROWTH Arabidopsis E3 ubiquitin ligases affect apical-basal PIN sorting decisions"

-Also in the abstract (line 41):

"Our findings reveal so far unknown principles of PIN polarity acquisition and establish a distinct function for E3 ligases, acting in the selective targeting of polarly localized plasma membrane protein cargo in higher plants."

- In the introduction (line 76):

"Genetic and cell biological analysis reveals so far unprecedented functions for E3 ubiquitin

ligases in the control of polar PM cargo sorting in general, and directional organ growth by defining PIN PM distribution in particular."

Finally, in the discussion (line 407):

"In particular, these E3s function in apical PIN sorting in cytokinetic root meristem cells, establishing WAV3/WAVH proteins as key determinants for directional auxin transport."

The authors have not shown sufficient evidence to support these claims. There is no evidence of E3 ubiquitin ligase activity directly controlling PIN polar cargo sorting. There are only indirect observations, as the mutant phenotypes could simply be attributed to an indirect result of losing E3 ubiquitin ligase activity. Additionally, there is no evidence of a general problem with sorting of all the PIN family members or all apical cargo, only with PIN2 and the ectopically expressed PIN1-GFP3. No other apical or basal sorted protein appears to be affected. This is not a minor phrasing issue, the authors need to soften their claims throughout the manuscript, or show evidence supporting these central claims. Taking into account the author's response pointing that after years of research they have not been able to produce such direct evidence, we suggest the following changes to soften the claims in the text:

Response: Thank you very much for this constructive feedback. We went through the sections and edited the revised manuscript.

-The authors altered the previous title, but the new title still needs revision for accuracy, the title could be changed to:

"WAVY GROWTH Arabidopsis E3 ubiquitin ligases activity affects apical PIN sorting decisions."

Response: This has been changed accordingly.

-Suggested changes to soften the claims in the abstract (line 41):

"Our findings reveal the involvement of E3 ligases in the selective targeting of apically localized PINs in higher plants."

Response: We have edited this section in accordance with this suggestion.

-Suggested changes to soften the claims at the end of the introduction (line 76):

"Genetic and cell biological analyses reveals a connection between E3 ubiquitin ligase function and the control of apical PIN cargo sorting, influencing directional auxin flow and root growth."

Response: This has been changed accordingly.

-Suggested changes for discussion (line 407):

“In particular, these E3s function in apical PIN sorting in cytokinetic root meristem cells, establishing WAV3/WAVH proteins as factors in directional auxin transport.”

Response: This has been changed accordingly.

Other issues that we found in the conclusions:

-In line 396 reads:

"Taken together, our observations can be summarized to propose a function for WAV3 in preventing PIN from entering BFA-sensitive ARF GEF basal sorting/recycling pathways, thereby enabling its correct positioning at apical PM domains."

Most of the evidence shown, however, indicates that the WAV3 function somehow prevents PIN from entering BFA-sensitive ARF-GEF secretory pathway. Similar to the authors' conclusion in line 292: "Collectively, these results indicate that WAV3/WAVH function predominantly in the sorting of newly synthesized PIN2 into the apical secretory pathway."

-we suggest changing to:

"Taken together, our results propose a function for WAV3 predominantly preventing PIN2 and PIN1-GFP3 entry into an ARF GEF basal sorting and secretion pathway".

Response: This has been changed accordingly.